# Beyond Synthetic Replays for Turning Diffusion Features into Incremental Knowledge

## Abstract

Few-shot class-incremental learning (FSCIL) is challenging due to extremely limited training data while requiring models to acquire new knowledge without catastrophic forgetting. Recent works have explored generative models, particularly Stable Diffusion (SD), to address these challenges. However, existing approaches use SD mainly as a replay generator, whereas we demonstrate that SD's rich multi-scale representations can serve as a unified backbone. Motivated by this observation, we introduce Diffusion-FSCIL, which extracts four synergistic feature types from SD by capturing real image characteristics through inversion, providing semantic diversity via class-conditioned synthesis, enhancing generalization through controlled noise injection, and enabling replay without image storage through generative features. Unlike conventional approaches requiring synthetic buffers and separate classification backbones, our unified framework operates entirely in the latent space with only lightweight networks ($\approx$6M parameters). Extensive experiments on CUB-200, miniImageNet, and CIFAR-100 demonstrate state-of-the-art performance, with comprehensive ablations confirming the necessity of each feature type. Furthermore, we confirm that our streamlined variant maintains competitive accuracy while substantially improving efficiency, establishing the viability of generative models as practical and effective backbones for FSCIL.

## 1 Introduction

Continual learning aims to enable models to acquire new knowledge sequentially while preserving previously learned information. However, real-world scenarios rarely provide abundant data for each new task, necessitating learning from limited samples. This practical constraint has led to the emergence of few-shot class-incremental learning (FSCIL), a challenging scenario where models incrementally acquire new classes from only a handful of samples while retaining prior knowledge. Unlike standard class-incremental learning (CIL), FSCIL must achieve effective generalization from minimal examples while simultaneously preventing catastrophic forgetting and overfitting. To address these challenges, researchers have developed diverse strategies (Chi et al., 2022; Akyürek et al., 2022; Zhang et al., 2021), with two primary directions emerging: enhancing backbone generalizability (Ahmed et al., 2024; Peng et al., 2022; Lee et al., 2025) and developing effective replay mechanisms (Agarwal et al., 2022; Liu et al., 2022; Shankarampeta & Yamauchi, 2021).

The first direction has increasingly focused on leveraging large-scale discriminative architectures as powerful backbones. With the advent of Vision Transformers (ViTs) (Oquab et al., 2023; Ilharco et al., 2021; Dosovitskiy et al., 2021) pre-trained on massive datasets (*e.g.,* ImageNet-21K (Russakovsky et al., 2015a)), these models have become the rising option. ViT-based methods (Wang et al., 2022; Park et al., 2024; Chen et al., 2025; Sun et al., 2024) have achieved strong performance; however, they often require extensive auxiliary resources such as large language models or explicit class-name prompts at inference, increasing complexity and limiting applicability. Moreover, because their discriminative features are already well-suited for FSCIL objectives, they may diminish the need to develop genuine few-shot adaptation capabilities and forgetting prevention mechanisms.

In parallel, the second direction has explored generative replay mechanisms for mitigating catastrophic forgetting. While early methods (Agarwal et al., 2022; Liu et al., 2022) primarily employed Generative Adversarial Networks (GANs) to synthesize replay images, recent studies (Kim et al., 2024; Jodelet et al., 2023; Meng et al., 2024; Jodelet et al., 2025) have adopted Stable Diffusion

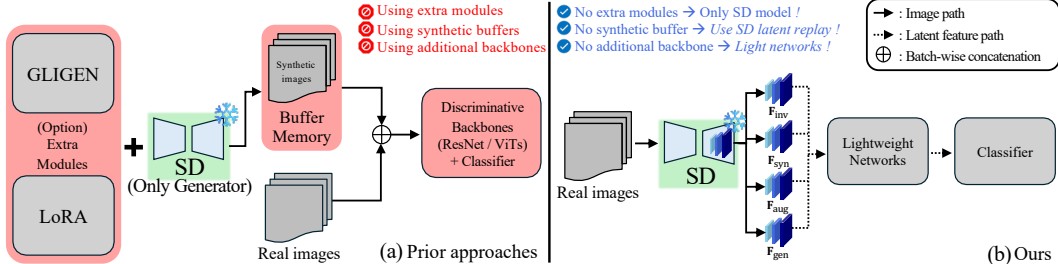

Figure 1: **Overview of Stable Diffusion (SD)-based continual learning pipelines. (a)** Prior approaches (Kim et al., 2024; Meng et al., 2024; Jodelet et al., 2023; 2025) employ SD only as a generator, often with extra modules (*e.g.,* GLIGEN, LoRA). Synthetic images must be stored in buffer memory and then passed, together with real images, to a separate discriminative backbone (ResNet/ViTs) for classification. **(b) Ours**: We directly use SD as both generator and frozen backbone, enabling extraction of multi-scale latent features (*e.g.,*, $\mathbf{F}_{inv}$, $\mathbf{F}_{syn}$, $\mathbf{F}_{aug}$, $\mathbf{F}_{gen}$) to handle FSCIL challenges through a unified framework. This eliminates the need for buffer memory, extra modules, and additional backbones, enabling training in the latent space with lightweight networks ($\approx$6M parameters). Black arrows indicate image/latent paths, and $\oplus$ denotes batch-wise concatenation.

(SD) (Rombach et al., 2022) for higher-quality generation. However, as illustrated in Fig. 1(a), these approaches relied on SD only as a generator. While effective when leveraging existing discriminative models, this design requires caching synthetic images in buffer memory and maintaining separate discriminative backbones. Such separation can introduce representational discrepancies between generator and classifier, often necessitating additional modules (*e.g.,* GLIGEN (Li et al., 2023), LoRA (Hu et al., 2022)) to ensure generation fidelity, thereby increasing operational complexity. Moreover, most works remained focused on standard CIL rather than FSCIL. Meanwhile, recent studies (Luo et al., 2024; Zhang et al., 2024b;a) have shown that SD's U-Net encodes semantically rich, multi-scale features exploitable for diverse tasks beyond image generation—though trained on web-scale data, these representations still require additional components for adaptation to be applied to discriminative recognition. Furthermore, SD's potential as a unified backbone for continual learning has not been explored, and its role in FSCIL remains absent.

To ground our new framework that leverages SD, we begin with a pilot study (Sec. 3.2) to examine two fundamental questions: (1) whether SD can serve as an effective backbone for FSCIL despite its generative training objective, and (2) whether conventional synthetic replay is sufficiently effective in this setting. Our findings provide clear motivation. First, SD, when used as a frozen feature extractor, exhibits surprisingly strong knowledge retention compared to large-scale discriminative models such as DINOv2 and OpenCLIP. Second, naïve synthetic replay progressively degrades performance—even with high-quality SD generations, adding more synthetic images counterintuitively worsens results. These results indicate that simple prompt-based replay fails to capture the training distribution and highlight the need for more sophisticated strategies to fully harness SD in FSCIL.

Building on these insights, we propose Diffusion-FSCIL, a framework that fully exploits SD as a unified backbone for FSCIL (Fig. 1(b)). Our key contributions include: (1) extracting complementary multi-scale features through both inversion and generation processes, (2) synthesizing class-specific latent features for replay without image storage, (3) employing controlled latent-space augmentation for better generalization, and (4) specialized training protocols that effectively leverage these diverse features. We maintain efficiency by freezing the SD backbone and training only lightweight components ($\approx$6M parameters). We achieve state-of-the-art performance on CUB-200, miniImageNet, and CIFAR-100, demonstrating that generative models can serve as competitive alternatives to discriminative backbones in extreme FSCIL scenarios.

## 2 RELATED WORK

**Few-shot class-incremental learning (FSCIL).** FSCIL extends class-incremental learning (CIL) to scenarios with extremely limited samples per new class, intensifying catastrophic forgetting and overfitting. Common strategies (Ahmed et al., 2024; Zhou et al., 2022; Peng et al., 2022; Lee et al., 2025; Song et al., 2023; Tang et al., 2024; Yang et al., 2023) introduce robust prototypes or feature representations to balance old and new classes, while another (Chi et al., 2022) approach adopts meta-learning to quickly adapt to novel few-shot categories. Many of these approaches further en-

hance backbone generalization through strong augmentation strategies such as CutMix (Yun et al., 2019), often combined with contrastive learning objectives to jointly improve semantic representations. In parallel, replay-based methods employ GAN-based generators (Liu et al., 2022; Agarwal et al., 2022; Shankarampeta & Yamauchi, 2021) to synthesize samples and alleviate forgetting. Although effective, these strategies remain constrained by their reliance on handcrafted augmentation or the limited fidelity of generated data, underscoring the need for more versatile solutions.

**Large-scale discriminative backbones.** Recent FSCIL research has shifted towards using large-scale discriminative backbones that already possess strong generalization ability due to pre-training. Beyond ResNets (He et al., 2016), Vision Transformers (ViTs) (Dosovitskiy et al., 2021; Oquab et al., 2023; Ilharco et al., 2021) pre-trained on massive datasets (*e.g.,* ImageNet-21K (Russakovsky et al., 2015a)) have become rising alternatives. Methods such as BiMC (Chen et al., 2025), PriVi-iLege (Park et al., 2024), L2P (Wang et al., 2022), and FineFMPL (Sun et al., 2024) have shown strong performance. However, these approaches often embed substantial discriminative knowledge before incremental training begins, reducing the need for genuine adaptation in FSCIL. Moreover, some methods (Park et al., 2024; Chen et al., 2025) rely on heavy auxiliary resources such as large language models, while others (Sun et al., 2024; Chen et al., 2025) depend on explicit class-name prompts at inference - both of which increase complexity and hinder practical deployment.

**Diffusion models and applications.** Diffusion models have recently been applied to continual learning mainly as replay generators to mitigate forgetting (Kim et al., 2024; Meng et al., 2024; Jodelet et al., 2023). Such approaches treat Stable Diffusion (SD) merely as an image generator, requiring buffer memory for storing synthetic images, separate discriminative backbones, or additional modules (*e.g.,* GLIGEN (Li et al., 2023), LoRA (Hu et al., 2022)) to improve quality. In parallel, recent works have demonstrated that SD encodes semantically rich multi-scale features that enable strong performance across downstream tasks such as segmentation (Marcos-Manchón et al., 2024) and correspondence matching (Kondapaneni et al., 2024). Yet, this representational capacity remains largely unexplored in FSCIL, where limited data poses significant challenges. In this paper, we explore SD not only as a generator but also as a unified backbone that contributes semantically rich features while maintaining efficiency via lightweight trainable networks.

## 3 BACKGROUND

This section briefly reviews preliminaries for few-shot class-incremental learning (FSCIL) and text-to-image (T2I) diffusion model, which are our problem setting and main component to constitute the overall pipeline, respectively. We then introduce our rationale to exploit the diffusion model in the context of FSCIL via the pilot study.

### 3.1 PRELIMINARY

**Few-shot class-incremental learning (FSCIL) setup.** We follow the standard FSCIL protocol (Yang et al., 2023; Ahmed et al., 2024), which incrementally trains a model across sequential sessions $\{\mathcal{S}^0, \mathcal{S}^1, \dots, \mathcal{S}^i\}$. In the base session ($\mathcal{S}^0$), the model is trained on a relatively large dataset containing base classes $\mathcal{C}^0$. Each incremental session $\mathcal{S}^i$ ($i \geq 1$) introduces new classes $\mathcal{C}^i$ with extremely limited samples per class (*e.g.,*, 5- or 10-shot). To mitigate catastrophic forgetting, a small exemplar memory (Zhao et al., 2023; Ahmed et al., 2024) is maintained. Evaluation is performed on all encountered classes up to session $i$, defined as $\mathcal{C}^{0:i} = \bigcup_{k=0}^{i} \mathcal{C}^k$.

**Text-to-image (T2I) diffusion model.** Stable Diffusion (SD) (Rombach et al., 2022) learns rich semantic representations from diverse image–text data through generative objectives. SD first encodes an image into latent $\mathbf{z}_0$ via a variational autoencoder (VAE) (Esser et al., 2021). During the *inversion* (forward) process, Gaussian noise is progressively added to $\mathbf{z}_0$ until reaching a fully noised latent $\mathbf{z}_T$. The U-Net $\boldsymbol{\epsilon}_\theta$ (Ronneberger et al., 2015) is trained to predict added noise at each timestep $t$, minimizing:

$$\mathcal{L}_{\text{SD}} = \mathbb{E}_{\mathbf{z}_t, t, \boldsymbol{\epsilon} \sim \mathcal{N}(0,1)} \big[ \| \boldsymbol{\epsilon} - \boldsymbol{\epsilon}_\theta(\mathbf{z}_t, t, \mathbf{w}) \|_2^2 \big], \tag{1}$$

where text embeddings $\mathbf{w}$ are obtained from text prompts using the text encoder (Radford et al., 2021). During the *generation* (reverse) process, the trained U-Net $\boldsymbol{\epsilon}_\theta$ iteratively denoises $\hat{\mathbf{z}}_T$ to $\hat{\mathbf{z}}_0$ under the guidance of text embeddings $\mathbf{w}$, yielding clean latents that produce synthetic images via VAE decoding. We use $\mathbf{z}$ for the inversion process and $\hat{\mathbf{z}}$ for the generation process.

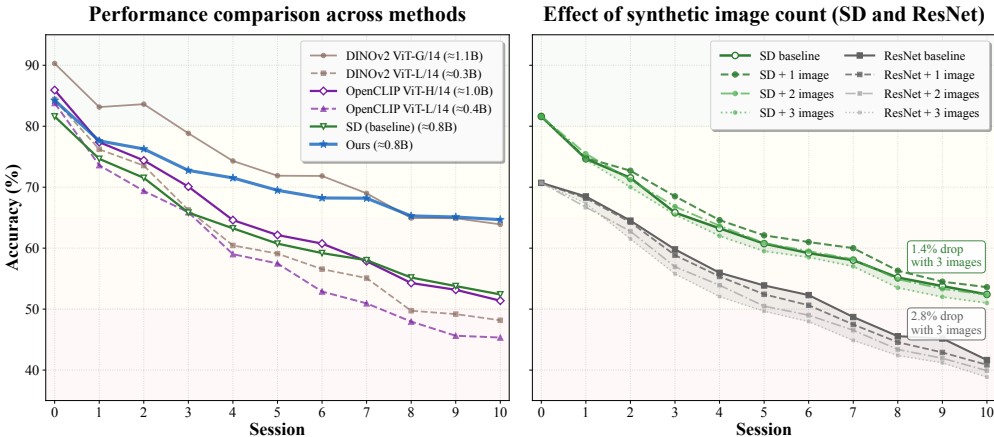

Figure 2: **Pilot study on using Stable Diffusion (SD) as backbone (left) and SD-generated replays (right).** **Left**: Comparison on CUB-200 using identical multi-layer feature extraction across large backbones (*e.g.,*, DINOv2, OpenCLIP, and SD, chosen to ensure billion-scale parameters); "Ours" denotes SD fully leveraged as our framework. Ours performs much better, and the baseline SD is also competitive. **Right**: Impact of the number of SD-generated images (1–3 per class) replays from class-name prompts for SD and ResNet training; naïvely using the synthetic images brings no gains.

## 3.2 PILOT STUDY

Before introducing our framework, we conduct a pilot study to address two fundamental questions:

**Can Stable Diffusion (SD) serve as a viable backbone for FSCIL?** Fig. 2 (left) compares SD with renowned large-scale ViTs such as DINOv2 (Oquab et al., 2023) and OpenCLIP (Ilharco et al., 2021) under identical multi-layer feature extraction on CUB-200. The SD baseline shows stronger retention than DINOv2-L and OpenCLIP-L/H despite its generative training objective. While DINOv2-G achieves higher initial accuracy due to its larger capacity, it rapidly overfits across sessions (more details in Appendix A). Our framework fully exploits SD's representational capacity to ultimately surpass even DINOv2-G by the final session, demonstrating that generative backbones can be competitive alternatives to discriminative ones in FSCIL.

**Are synthetic replays generated with SD sufficient for FSCIL?** A straightforward use of SD is to generate replay samples for preventing forgetting. Recent works (Kim et al., 2024; Meng et al., 2024; Jodelet et al., 2023; 2025) actually explored SD for replay to mitigate forgetting, yet mostly by directly generating synthetic images. Fig. 2 (right) examines this approach's effectiveness by replaying 1–3 synthetic images per class from class-name prompts using both ResNet and SD baselines. For ResNet, replay variations consistently degrade performance; though SD maintains accuracy with a single image, adding more images also causes degradation. This indicates that generated replays from SD do not work for capturing the training distribution, consistent with Kim et al. (2025), showing simple prompt-based replay generation is insufficient. Therefore, we believe that SD requires careful strategies to harness its potential for FSCIL.

## 4 METHOD

This section proposes a novel few-shot class-incremental learning (FSCIL) framework that leverages SD's rich representational capacity as a unified frozen backbone. Our approach includes four components: extracting complementary features from diffusion processes (Sec. 4.1); class-specific generative replay without synthetic image storage (Sec. 4.2); latent-space augmentation for limited samples (Sec. 4.3); and specialized training protocols (Sec. 4.4). The overview is illustrated in Fig. 3.

### 4.1 EXTRACTING INVERSION AND SYNTHETIC FEATURES

The U-Net of SD inherently produces multi-scale features that encompass both detailed patterns and higher-level semantics. Prior study (Luo et al., 2024) shows that these representations vary in resolution and abstraction, providing diverse information from local detail to global semantic structure. Such diversity is particularly important for FSCIL, which benefits from representations that combine fine detail and semantic generalization. In practice, we exploit intermediate representations

Figure 3: **Schematic overview of our framework.** An input image (*e.g.,*, Herring Gull) is encoded into latent $\mathbf{z}_0$, then transformed into different noise levels: $\hat{\mathbf{z}}_1$ (noised latent via DDIM scheduler), $\hat{\mathbf{z}}_t$ (partially noised at $t \in (1, T]$), and pure noise $\hat{\mathbf{z}}_T$. Stable Diffusion's frozen U-Net extracts multi-scale features under different conditions: inversion feature $\mathbf{F}_{\text{inv}}$ from $\mathbf{z}_0$ with null prompt, synthetic feature $\mathbf{F}_{\text{syn}}$ from $\hat{\mathbf{z}}_1$ with class-name prompt, augmented feature $\mathbf{F}_{\text{aug}}$ from $\hat{\mathbf{z}}_t$ via partial generation with class-specific prompt, and generative feature $\mathbf{F}_{\text{gen}}$ from $\hat{\mathbf{z}}_T$ via full generation with class-specific prompt (incremental sessions only). All features are aggregated and passed through lightweight networks ($g^{\text{agg}}$, $g^{\text{conv}}$, $g^{\text{MLP}}$ - about 6M parameters, details in Appendix B) and the prototype-based classifier (Yang et al., 2023), while SD remains frozen.

from U-Net layers 4–12, which strike a balance between detail and abstraction, while excluding the lowest-resolution 1–3 layers that we empirically confirm to be less informative (see Appendix B).

**One-step inversion feature.** Given a latent variable $\mathbf{z}_0$ from the VAE encoder, we feed $\mathbf{z}_0$ into SD's U-Net with a null prompt to obtain multi-scale features $\{\mathbf{f}_4, \dots, \mathbf{f}_{12}\}$. These features are passed through the aggregation network $g^{\text{agg}}$, which estimates adaptive weights $\beta_l$ for each layer $l$. The coefficients $\beta_l$ act as importance weights, enabling $g^{\text{agg}}$ to emphasize informative scales while down-weighting less useful ones (detailed architecture is provided in Appendix B). The inversion feature is then defined as $\mathbf{F}_{\text{inv}} = \sum_{l=4}^{12} \beta_l \mathbf{f}_l$. In addition, the deterministic DDIM scheduler (Song et al., 2020) naturally yields the noised latent $\hat{\mathbf{z}}_1$, which serves as input for subsequent feature extraction.

**One-step synthetic feature.** Starting from the noised latent $\hat{\mathbf{z}}_1$, we exploit SD's generation capability by conditioning the U-Net on a class-name prompt $\mathbf{p}_c$ (*e.g.,*, "Herring_Gull"). Multi-scale features $\{\mathbf{f}'_4, \dots, \mathbf{f}'_{12}\}$ are extracted from the same U-Net and passed through $g^{\text{agg}}$. As before, $g^{\text{agg}}$ estimates weights $\beta_l$ to combine scales adaptively, producing the synthetic feature $\mathbf{F}_{\text{syn}} = \sum_{l=4}^{12} \beta_l \mathbf{f}'_l$, where $\beta_l$ are same weights used for $\mathbf{F}_{\text{inv}}$ and are shared across feature generation. This design ensures both features share a consistent multi-scale basis, while prompt conditioning introduces semantic diversity that enriches the original characteristics preserved in $\mathbf{F}_{\text{inv}}$.

**Discussion on diffusion steps.** For both efficiency and feature quality, we adopt a one-step strategy within the diffusion timestep range. Intuitively, extracting features across multiple timesteps in the diffusion process is redundant and often yields semantically degraded representations, since larger timesteps correspond to latents that are increasingly dominated by noise (see Appendix E). Prior works (Kondapaneni et al., 2024; Wang et al., 2024) likewise show that minimally perturbed latents yield the most informative representations. We thus use the minimal timestep (one-step) to extract both features ($\mathbf{F}_{\text{inv}}$, $\mathbf{F}_{\text{syn}}$) consistently during training.

## 4.2 TEXT-BASED CLASS-SPECIFIC GENERATIVE FEATURE

Building upon our feature extraction framework, we extend this paradigm to enable generative replay without storing synthetic images. Specifically, we leverage SD's full generation process starting from pure noise $\hat{\mathbf{z}}_T$ and use class-name prompts $\mathbf{p}$ (*e.g., "a photo of {class-name}"*) to guide the generation process toward target class representations. At the final timestep, we extract the generative feature $\mathbf{F}_{\text{gen}}$ following the same aggregation procedure as $\mathbf{F}_{\text{syn}}$ from the fully denoised latent $\hat{\mathbf{z}}_0$, enabling latent-space generative replay without input images. However, when guided by class-name prompts $\mathbf{p}$, the generative feature often fails to preserve subtle, class-specific details (see Appendix C).

To address this limitation, we define a class-specific prompt $\mathbf{p}_c^*$ for each class $c$, represented by a learnable embedding $\mathbf{w}_c^*$ introduced into CLIP's text embedding space, inspired by Gal et al. (2022). Each $\mathbf{w}_c^*$ is initialized from the corresponding class name (*e.g.,* "cardinal", "Herring_Gull") and optimized to encode fine-grained class semantics using SD loss (Eq. 1):

$$\mathbf{w}_c^* = \arg\min_{\mathbf{w}_c} \mathbb{E}_{\mathbf{z}_t, t, \boldsymbol{\epsilon} \sim \mathcal{N}(0,1)} \left[ \| \boldsymbol{\epsilon} - \boldsymbol{\epsilon}_\theta(\mathbf{z}_t, t, \tau_\theta(\mathbf{p}_c^*)) \|_2^2 \right], \tag{2}$$

where both the U-Net $\boldsymbol{\epsilon}_\theta$ and the text encoder $\tau_\theta$ remain frozen. After that, the optimized embedding $\mathbf{w}_c^*$ captures fine-grained class semantics. During generation, the class-specific prompts $\mathbf{p}^*$ is used to call up the textual embeddings $\mathbf{w}^*$ to guide the generation process, producing the generative feature $\mathbf{F}_{\text{gen}}$ that effectively retains previously learned knowledge while serving as latent replay, requiring neither buffer memory for synthetic images nor auxiliary modules to enhance image fidelity.

## 4.3 AUGMENTED FEATURE VIA LATENT-SPACE VARIATION

FSCIL's fundamental challenge lies in the scarcity of training samples per class. A straightforward way to address this is to generate additional samples using class-specific prompts $\mathbf{p}^*$ for the newly introduced classes. However, this prompt-based generation faces two limitations. Samples generated from pure noise ($\hat{\mathbf{z}}_T \to \hat{\mathbf{z}}_0$) may capture general class semantics but often deviate from the specific characteristics present in the few training examples. Moreover, such samples are not guaranteed to remain aligned with the original data distribution, which can introduce harmful shifts and degrade incremental performance.

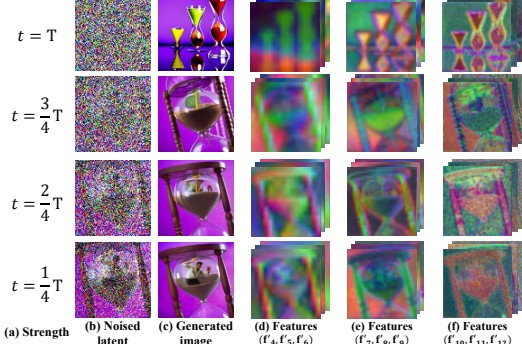

Figure 4: **Visualizations of augmented features $\mathbf{F}_{\text{aug}}$.** (a) Noise strengths $t$, (b) noise-injected latents, (c) denoised outputs (visualization only), and (d–f) extracted multi-scale features $\mathbf{f}_l'$ in the generation process.

To overcome these issues, we propose a latent-space augmentation strategy that introduces controlled variations while remaining anchored to the original samples. Given an image latent $\mathbf{z}_0$, we inject Gaussian noise at a specific timestep $t \in (1, T]$, yielding a partially noised latent $\hat{\mathbf{z}}_t$. We then apply partial generation ($\hat{\mathbf{z}}_t \to \hat{\mathbf{z}}_0$) and extract the augmented feature $\mathbf{F}_{\text{aug}}$ following the same aggregation procedure as $\mathbf{F}_{\text{syn}}$ from the final latent $\hat{\mathbf{z}}_0$. This preserves the structural foundation of the original samples while introducing variation through controlled noise injection.

**Balancing fidelity and diversity.** The strength of the injected noise, controlled by the choice of timestep $t$, directly determines the extent of variation. Larger timesteps (close to T) produce highly diverse but less faithful samples, whereas smaller timesteps (close to 0) preserve fidelity of the original sample but yield limited diversity. To balance this trade-off, we discretize the timestep range into $m$ segments and randomly select $t \in \left\{ \frac{T}{m}, \frac{2T}{m}, \ldots, \frac{(m-1)T}{m}, T \right\}$, where $m > 1$. In Fig. 4, we show variations of $\mathbf{F}_{\text{aug}}$ across noise strengths, highlighting the differences.

Importantly, when $t = T$, the augmented feature $\mathbf{F}_{\text{aug}}$ becomes identical to the full generation case $\mathbf{F}_{\text{gen}}$. We therefore separate their roles: $\mathbf{F}_{\text{gen}}$ is used exclusively for replay of previously learned classes, ensuring knowledge retention against forgetting, while $\mathbf{F}_{\text{aug}}$ is applied only to newly introduced classes in the current session, enhancing generalization.

## 4.4 TRAINING AND INFERENCE PROTOCOLS

**Base session ($\mathcal{S}^0$).** We utilize three feature types ($\mathbf{F}_{\text{inv}}$, $\mathbf{F}_{\text{syn}}$, and $\mathbf{F}_{\text{aug}}$) during base session training with frozen SD. The aggregation network $g^{\text{agg}}$, the convolution layer $g^{\text{conv}}$, and the MLP layer $g^{\text{MLP}}$ are all trained in this stage. To improve generalization while considering computational cost, $\mathbf{F}_{\text{aug}}$ is employed only during the final training epochs as a brief fine-tuning step.

**Incremental sessions ($\mathcal{S}^{i \geq 1}$).** During incremental sessions, only the MLP layer $g^{\text{MLP}}$ are trained with limited data $\mathcal{D}^i$, while the other components remain frozen. Four feature types ($\mathbf{F}_{\text{inv}}$, $\mathbf{F}_{\text{syn}}$, $\mathbf{F}_{\text{aug}}$, and $\mathbf{F}_{\text{gen}}$) are employed in this stage, with $\mathbf{F}_{\text{gen}}$ being introduced starting from here to serve as latent replay for mitigating catastrophic forgetting.

Table 1: **FSCIL results on CUB-200**. Top-1 accuracies (%) for sessions 0–10 (0 = base session). AA: average accuracy; FI: final improvement over previous methods. [†]Results from Yang et al. (2023). Methods under *controlled SD backbone* use the same SD backbone as ours. **Bold**=best; underline=second best per column.

| Methods | Session Acc. (%) | | | | | | | | | | | AA (%) | FI |
| | 0 | 1 | 2 | 3 | 4 | 5 | 6 | 7 | 8 | 9 | 10 | | |
|---|---|---|---|---|---|---|---|---|---|---|---|---|---|
| *Methods that used their respective backbone (e.g., ResNet-18)* | | | | | | | | | | | | | |
| TOPIC[†] (Tao et al., 2020) | 68.7 | 62.5 | 54.8 | 50.0 | 45.3 | 41.4 | 38.4 | 35.4 | 32.2 | 28.3 | 26.3 | 43.9 | +44.0 |
| CEC[†] (Zhang et al., 2021) | 75.9 | 71.9 | 68.5 | 63.5 | 62.4 | 58.3 | 57.7 | 55.8 | 54.8 | 53.5 | 52.3 | 61.3 | +18.0 |
| FACT[†] (Zhou et al., 2022) | 75.9 | 73.2 | 70.8 | 66.1 | 65.6 | 62.2 | 61.7 | 59.8 | 58.4 | 57.9 | 56.9 | 64.4 | +13.4 |
| ALICE[†] (Peng et al., 2022) | 77.4 | 72.7 | 70.6 | 67.2 | 65.9 | 63.4 | 62.9 | 61.9 | 60.5 | 60.6 | 60.1 | 65.7 | +10.2 |
| ERDFR (Liu et al., 2022) | 75.9 | 72.1 | 68.6 | 63.8 | 62.6 | 59.1 | 57.8 | 55.9 | 54.9 | 53.6 | 52.4 | 61.5 | +17.9 |
| NC-FSCIL[†] (Yang et al., 2023) | 80.5 | 76.0 | 72.3 | 70.3 | 68.2 | 65.2 | 64.4 | 63.3 | 60.7 | 60.0 | 59.4 | 67.3 | +10.9 |
| BiDistFSCIL (Zhao et al., 2023) | 79.1 | 75.4 | 72.8 | 69.1 | 67.5 | 65.1 | 64.0 | 63.5 | 61.9 | 61.5 | 60.9 | 67.3 | +9.4 |
| SAVC (Song et al., 2023) | 81.9 | 77.9 | 75.0 | 70.2 | 70.0 | 67.0 | 66.2 | 65.3 | 63.8 | 63.2 | 62.5 | 69.4 | +7.8 |
| OrCo (Ahmed et al., 2024) | 75.6 | 66.8 | 64.1 | 63.7 | 62.2 | 60.3 | 60.2 | 59.2 | 58.0 | 54.9 | 52.1 | 61.6 | +18.2 |
| CLOSER (Oh et al., 2024) | 79.4 | 75.9 | 73.5 | 70.5 | 69.2 | 67.2 | 66.7 | 65.7 | 64.0 | 64.0 | 63.6 | 69.1 | +6.7 |
| Yourself (Tang et al., 2024) | 83.4 | 77.0 | 75.3 | 72.2 | 69.0 | 66.8 | 66.0 | 65.6 | 64.1 | 64.5 | 63.6 | 69.8 | +6.7 |
| Tri-WE (Lee et al., 2025) | 81.6 | 78.6 | 76.1 | 73.6 | 71.8 | 69.1 | 67.8 | 66.8 | 65.8 | 65.0 | 63.9 | 70.9 | +6.4 |
| *Methods using a fixed SD backbone (controlled for fair comparison with ours)* | | | | | | | | | | | | | |
| SDDR (Jodelet et al., 2023) | 86.5 | 80.2 | 78.1 | 74.1 | 71.2 | 68.4 | 66.6 | 66.1 | 62.4 | 61.4 | 60.1 | 70.5 | +10.2 |
| NC-FSCIL (Yang et al., 2023) | 86.5 | 79.9 | 78.2 | 73.8 | 70.4 | 68.5 | 67.7 | 66.1 | 63.2 | 62.7 | 61.1 | 70.7 | +9.2 |
| Diff-Class (Meng et al., 2024) | 86.5 | 78.2 | 75.9 | 70.0 | 66.5 | 65.0 | 63.2 | 62.0 | 58.4 | 57.6 | 56.1 | 67.2 | +14.2 |
| COMP-FSCIL (Zou et al., 2024) | 84.6 | 81.0 | 77.5 | 73.3 | 70.7 | 68.0 | 66.4 | 64.9 | 62.9 | 62.4 | 61.2 | 70.3 | +9.1 |
| OrCo (Ahmed et al., 2024) | 81.1 | 71.2 | 68.1 | 64.4 | 58.3 | 57.0 | 55.0 | 52.2 | 48.5 | 46.4 | 45.7 | 58.9 | +24.6 |
| CLOSER (Oh et al., 2024) | 86.5 | 80.2 | 77.8 | 74.5 | 72.8 | 70.5 | 69.4 | 68.6 | 66.8 | 66.6 | 65.7 | 72.7 | +4.6 |
| **Ours** | **86.6** | **81.3** | **80.1** | **77.7** | **75.9** | **74.2** | **72.4** | **72.5** | **70.8** | **70.4** | **70.3** | **75.7** | |

**Training loss.** We adopt the prototype-based classifier with dot-regression (DR) loss $\mathcal{L}_{\text{DR}}$, suggested by Yang et al. (2023), for three feature types ($\mathbf{F}_{\text{inv}}$, $\mathbf{F}_{\text{syn}}$, and $\mathbf{F}_{\text{aug}}$). For the generative feature $\mathbf{F}_{\text{gen}}$, we employ a lightweight MLP-based knowledge distillation strategy to mitigate potential misalignment with real training samples (*i.e.,* generative bias), similarly to (Smith et al., 2021; Kim et al., 2024), transferring knowledge from a frozen teacher model $\phi_{\text{t}}(\cdot)$ to a student model $\phi_{\text{s}}(\cdot)$ by minimizing cosine distance: $\mathcal{L}_{\text{distill}} = 1 - \cos\left(\phi_{\text{t}}(\mathbf{F}_{\text{gen}}), \phi_{\text{s}}(\mathbf{F}_{\text{gen}})\right)$. Our overall loss for the incremental sessions is $\mathcal{L}_{\text{total}}^{i} = \mathcal{L}_{\text{DR}} + \gamma^{i}\mathcal{L}_{\text{distill}}$, where $\gamma^{i} = \gamma_{\text{init}} + \frac{i}{\mathcal{S}_{\text{total}}}(1 - \gamma_{\text{init}})$ linearly increases across sessions to address growing forgetting. Here, $\gamma_{\text{init}}$ is the initial coefficient, $i$ the current session, and $\mathcal{S}_{\text{total}}$ the total number of sessions.

**Inference.** Since text prompts are unavailable during inference, we exclusively use the inversion feature $\mathbf{F}_{\text{inv}}$ extracted from image inputs with a "null" prompt, except for the CUB-200 where a "bird" prompt is used due to dataset specificity.

## 5 EXPERIMENT

### 5.1 SETUP

**Implementation details.** We evaluate on CUB-200 (Wah et al., 2011), miniImageNet (Russakovsky et al., 2015b), and CIFAR-100 (Krizhevsky et al., 2009). Following standard protocol (Zhang et al., 2021; Yang et al., 2023): CUB-200 uses 100 base classes with 10 incremental sessions (10-way, 5-shot); miniImageNet and CIFAR-100 use 60 base classes with 8 incremental sessions (5-way, 5-shot). We adopt pre-trained Stable Diffusion v1.5 (Rombach et al., 2022) with classifier-free guidance scale 7.5 (Ho & Salimans, 2022). Images are resized to $512 \times 512$; one sample per class is stored following (Zhao et al., 2023; Ahmed et al., 2024). Additional details in the Appendix B.

**Evaluation Metrics.** We employ standard FSCIL metrics: session accuracy at each incremental session $\mathcal{S}^{i}$; average accuracy (AA.) across all sessions; final accuracy improvement (FI) over competing methods; and final session accuracy (Acc.) on all classes. For ablation, we report Base. (accuracy on base classes $\mathcal{C}^{0}$) and Inc. (accuracy on incremental classes $\mathcal{C}^{i \geq 1}$) after the last session.

### 5.2 COMPARISON WITH OTHER METHODS

Tables 1 and 2 summarize our results on CUB-200 and miniImageNet, respectively, compared against state-of-the-art FSCIL methods. To ensure fair comparison, we consider two categories: (1) methods using their originally proposed backbones (*e.g.,* ResNet-18), and (2) methods adapted

Table 2: **FSCIL results on miniImageNet**. Top-1 accuracies (%) for sessions 0–8 (0 = base session). AA. denotes average accuracy across all sessions. FI calculates the improvement of our method in the last session compared to previous methods. [†]results from Yang et al. (2023). Methods under *controlled SD backbone* use the same SD backbone as ours. **Bold**=best; underline=second best per column.

| Methods | Session Acc. (%) | | | | | | | | | AA. (%) | FI |
|---|---|---|---|---|---|---|---|---|---|---|---|
| | 0 | 1 | 2 | 3 | 4 | 5 | 6 | 7 | 8 | | |
| *Methods that used their respective backbone (e.g., ResNet-18)* | | | | | | | | | | | |
| TOPIC[†] (Tao et al., 2020) | 61.3 | 50.1 | 45.2 | 41.2 | 37.5 | 35.5 | 32.2 | 29.5 | 24.4 | 39.6 | +40.3 |
| CEC[†] (Zhang et al., 2021) | 72.0 | 66.8 | 63.0 | 59.4 | 56.7 | 53.7 | 51.2 | 49.2 | 47.6 | 57.7 | +17.1 |
| FACT (Zhou et al., 2022) | 72.6 | 69.6 | 66.4 | 62.8 | 60.6 | 57.3 | 54.3 | 52.2 | 50.5 | 60.7 | +14.2 |
| ERDFR (Liu et al., 2022) | 71.8 | 67.1 | 63.2 | 59.8 | 57.0 | 54.0 | 51.6 | 49.5 | 48.2 | 58.0 | +16.5 |
| ALICE[†] (Peng et al., 2022) | 80.6 | 70.6 | 67.4 | 64.5 | 62.5 | 60.0 | 57.8 | 56.8 | 55.7 | 64.0 | +9.0 |
| NC-FSCIL[†] (Yang et al., 2023) | 84.0 | 76.8 | 72.0 | 67.8 | 66.4 | 64.0 | 61.5 | 59.5 | 58.3 | 67.8 | +6.4 |
| BiDistFSCIL (Zhao et al., 2023) | 74.7 | 70.4 | 66.3 | 62.8 | 60.8 | 57.2 | 54.8 | 53.7 | 52.2 | 61.4 | +12.5 |
| SAVC (Song et al., 2023) | 81.1 | 76.1 | 72.4 | 68.9 | 66.5 | 63.0 | 59.9 | 58.4 | 57.1 | 67.0 | +7.6 |
| OrCo (Ahmed et al., 2024) | 83.3 | 75.3 | 71.5 | 68.2 | 65.6 | 63.1 | 60.2 | 58.8 | 58.1 | 67.1 | +6.6 |
| CLOSER (Oh et al., 2024) | 76.0 | 71.6 | 68.0 | 64.7 | 61.7 | 58.9 | 56.2 | 54.5 | 53.3 | 62.8 | +11.4 |
| Yourself (Tang et al., 2024) | 84.0 | 77.6 | 73.7 | 70.0 | 68.0 | 64.9 | 62.1 | 59.8 | 59.0 | 68.8 | +5.7 |
| Tri-we (Lee et al., 2025) | 84.1 | 81.4 | 76.7 | 73.6 | 70.1 | 65.1 | 63.4 | 61.0 | 60.1 | 70.6 | +4.6 |
| *Methods using a fixed SD backbone (controlled for fair comparison with ours)* | | | | | | | | | | | |
| SDDR (Jodelet et al., 2023) | **89.7** | 83.0 | 78.3 | 74.4 | 71.0 | 66.9 | 63.5 | 61.0 | 60.1 | 72.0 | +4.6 |
| NC-FSCIL (Yang et al., 2023) | **89.7** | 83.0 | 77.9 | 73.4 | 70.7 | 66.0 | 62.0 | 59.8 | 58.9 | 71.3 | +5.8 |
| Diff-Class (Meng et al., 2024) | **89.7** | 83.2 | 78.2 | 74.1 | 71.3 | 68.1 | 65.1 | 63.3 | 61.8 | 72.8 | +2.9 |
| COMP-FSCIL (Zou et al., 2024) | 88.7 | 83.5 | **79.3** | 75.2 | 71.4 | 67.6 | 64.7 | 61.8 | 59.2 | 72.4 | +5.5 |
| CLOSER (Oh et al., 2024) | **89.7** | **83.7** | 78.7 | 76.2 | 73.0 | 70.0 | 67.0 | 65.0 | 63.6 | 74.1 | +1.1 |
| **Ours** | **89.7** | 82.3 | 78.8 | 75.3 | **73.2** | **70.5** | **67.6** | **65.8** | **64.7** | **74.2** | |

Table 3: (Left) Ablation study results on miniImageNet and CUB-200 datasets. (Right) Ablation study of the initial distillation value $\gamma_{init}$ on miniImageNet. $\gamma^i = 0$ denotes the case without distillation. In both tables, the gray row indicates our choice and **Bold** indicates the best results.

| Methods | miniImageNet | | | | CUB-200 | | | |
|---|---|---|---|---|---|---|---|---|
| | AA. | Acc. | Base. | Inc. | AA. | Acc. | Base. | Inc. |
| (a) $\mathbf{F}_{inv}$ | 70.5 | 58.9 | 82.0 | 24.3 | 71.2 | 60.3 | 75.6 | 45.4 |
| (b) (a) + $\mathbf{F}_{syn}$ | 70.4 | 59.6 | 79.3 | 29.9 | 70.7 | 61.0 | 75.9 | 46.4 |
| (c) (b) + $\mathbf{F}_{gen}$ | 72.5 | 61.5 | **82.7** | 29.8 | 74.1 | 67.1 | **82.7** | 51.8 |
| (d) (c) + $\mathbf{F}_{aug}$ | **73.4** | **64.7** | 80.4 | **41.2** | **74.9** | **70.3** | 79.6 | **61.2** |

| $\gamma_{init}$ | AA. | Acc. | Base. | Inc. |
|---|---|---|---|---|
| $\gamma^i = 0$ | 71.5 | 60.2 | 77.9 | 33.7 |
| 0.0 | 72.8 | 64.1 | 79.6 | 41.0 |
| 0.1 | 73.4 | **64.7** | 80.4 | **41.2** |
| 0.5 | 73.4 | 64.0 | 80.9 | 38.5 |
| 0.7 | **73.5** | 64.0 | **81.3** | 38.1 |

to use the same SD backbone as ours for controlled evaluation. We exclude large-scale discriminative approaches such as BiMC (Chen et al., 2025), FineFMPL (Sun et al., 2024), PriViLege (Park et al., 2024), and L2P (Wang et al., 2022), which rely on large-scale ViT backbones trained on datasets closely related to FSCIL benchmarks or explicit class-name information at inference.

Our method achieves state-of-the-art performance across all benchmarks. On CUB-200 (Tab. 1), we obtain the highest AA. (75.7%) and final Acc. (70.3%), significantly outperforming all methods regardless of backbone architecture—both controlled SD-backbone methods like CLOSER (+3.0% AA., +4.6% FI) and COMP-FSCIL (+5.4% AA., +9.1% FI), and original backbone methods like Tri-WE (+4.8% AA., +6.4% FI). On miniImageNet (Tab. 2), we reach 74.2% AA. and 64.7% Acc. Although CLOSER achieves a comparable AA. (74.1%), our method substantially outperforms it in final accuracy (+1.1% FI) while surpassing all other methods. We further evaluate on CIFAR-100 (Appendix Tab. B), where we achieve the best final Acc. (60.6%) and tie with Tri-WE for the highest AA. (68.7%). Tri-WE achieves slightly stronger accuracy in the early sessions, but it degrades more rapidly as sessions progress, whereas our method maintains higher accuracy in later stages.

## 5.3 ABLATION STUDY

We conduct comprehensive ablation studies to validate the effectiveness of our key components.

**On main components.** Tab. 3 (Left) presents ablation results on miniImageNet and CUB-200. Adding $\mathbf{F}_{syn}$ to the baseline inversion feature improves Inc., showing its effectiveness for adapting to new classes. Introducing $\mathbf{F}_{gen}$ yields substantial gains in AA. and Acc. while achieving the highest Base., confirming effective forgetting mitigation. Finally, $\mathbf{F}_{aug}$ provides the best overall results with dramatic Inc. improvements, showing enhanced generalization on new classes. Although Base. decreases slightly, this represents a favorable trade-off. These results demonstrate that all features work harmoniously together to ensure optimal FSCIL performance.

**On progressive distillation.** Tab. 3 (Right) investigates the impact of initial distillation value $\gamma_{\text{init}}$. Comparing $\gamma^i=0$ (no distillation) with distillation variants shows that $\mathcal{L}_{\text{distill}}$ is essential, improving AA. from 71.5% to over 72.8%. Among distillation options, while AA. remains similar (73.4–73.5%), Inc. varies dramatically. Higher values ($\gamma_{\text{init}} \in \{0.5, 0.7\}$) preserve Base. well but degrade few-shot samples, while $\gamma_{\text{init}} = 0.1$ provides the best Inc. (41.2%) with competitive Base. (80.4%). This demonstrates that excessive distillation strength hinders incremental learning by blocking new class information integration. Thus, we select $\gamma_{\text{init}} = 0.1$ for optimal balance between knowledge retention and adaptation.

Table 4: Ablation of diffusion noise interval $m$ on miniImageNet. The gray row indicates our choice ($m=4$).

Table 5: Ablation of single noise strength ($m=1$) on miniImageNet. The row *multi* corresponds to the multi-interval setting, and the gray row indicates our choice ($m=4$).

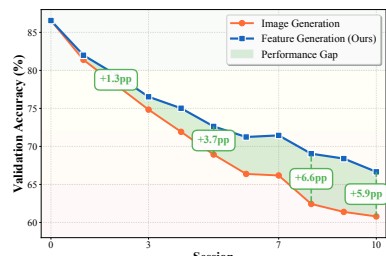

Figure 5: **Latent vs. image replay on CUB-200.** Performance gap is reported in percentage points (pp).

| Multi-interval noise strengths | | | | Single noise strength ($m=1$) | | | | |
|---|---|---|---|---|---|---|---|---|
| $m$ | AA. | Acc. | Base. | Inc. | $t$ | AA. | Acc. | Base. | Inc. |
| 2 | **73.4** | 64.6 | 79.8 | **41.9** | multi | **73.4** | **64.7** | 80.4 | 41.2 |
| 4 | **73.4** | **64.7** | 80.4 | 41.2 | 0.3T | 72.7 | 61.9 | **83.2** | 30.0 |
| 6 | **73.4** | 64.5 | **80.9** | 39.8 | 0.5T | 73.2 | 63.6 | 81.5 | 36.6 |
| | | | | | 0.7T | 73.1 | 64.1 | 78.6 | **42.3** |

**On noise variants.** Tab. 4 and Tab. 5 examine noise strategies for $\mathbf{F}_{\text{aug}}$. Multi-interval sampling consistently achieves superior performance across all values, with $m=4$ selected for optimal balance. In contrast, single-noise settings show the expected fidelity-diversity trade-off: lower noise levels (0.3T) enhance fidelity, strengthening Base., while higher noise levels (0.7T) increase diversity, improving Inc. but at the cost of base class retention. However, none of the single-noise configurations achieve balanced improvements across all metrics. These results demonstrate that performing latent-space augmentation with multi-interval sampling is crucial for balancing fidelity and diversity, with $m=4$ providing the most effective trade-off.

**Effectiveness of latent-space generative replay.** Fig. 5 compares synthetic image replay with our latent replay ($\mathbf{F}_{\text{gen}}$). Our approach consistently outperforms image replay across all sessions, with performance gaps progressively widening from +1.3pp in early sessions to +5.9pp in the final session. This demonstrates that latent replay preserves knowledge more effectively than synthetic images while eliminating storage overhead.

### 5.4 COMPUTATIONAL COSTS - TRAINING TIME

One may question the training cost of our method using SD compared to recent approaches in practice. Tab. 6 reports training time versus Acc. across incremental sessions on CUB-200 compared with Yourself (Tang et al., 2024). Our method inherently involves complete generative processes ($\mathbf{F}_{\text{gen}}$ and $\mathbf{F}_{\text{aug}}$), potentially increasing training costs compared to purely discriminative approaches. We introduce an efficient variant (dubbed *eff.ver*), which restricts the generative features used in $\mathbf{F}_{\text{gen}}$ and $\mathbf{F}_{\text{aug}}$, along with reducing the total training iterations

Table 6: Training time (TT) and final accuracy (Acc.) comparison on CUB-200.

| Model | TT (min) | Acc. (%) |
|---|---|---|
| Yourself | $\approx 1236$ | 63.6 |
| Ours (*eff.* ver.) | $\approx$**161** | 63.6 |
| Ours | $\approx 2070$ | **70.2** |

minimally. Compared to the recent SOTA method Yourself, our efficient variant achieves same Acc. (63.6%) while being approximately $7.7\times$ faster in training time. This result confirms that our framework effectively balances accuracy and training efficiency, offering flexibility based on available computational resources.

## 6 CONCLUSION

In this work, we revisited the role of diffusion models in few-shot class-incremental learning (FS-CIL) and highlighted the untapped potential of Stable Diffusion (SD) as a competitive backbone. Building on this insight, we introduced Diffusion-FSCIL, which exploits four complementary feature types—inversion, synthetic, augmented, and generative—extracted from SD's multi-scale representations to address limited data and catastrophic forgetting. Our framework achieves state-of-the-art performance on FSCIL benchmarks while maintaining efficiency, demonstrating that diffusion models can function not only as powerful generators but also as effective backbones for continual recognition. Further limitations and future directions are discussed in Appendix I.

# STATEMENTS

**Ethics Statement.** Our research does **NOT** conduct any kind of experiment that has potential issues such as human subjects, public health, privacy, fairness, security, etc. All authors confirm that they adhere to the ICLR Code of Ethics.

**Reproducibility Statement.** All datasets used in this paper, CUB-200, miniImageNet, and CIFAR-100, are publicly available and properly cited. We provide detailed descriptions of our experimental settings in the main paper (Sec. 5) and Appendix B, including hyper-parameters, dataset splits, architecture details, and prompt optimization procedures.

**LLM Usage Statement.** We used large language models solely as assistive tools for writing clearance, including grammar correction, style refinement, and minor wording adjustments. LLMs were not used for research ideation, experiment design, data analysis, result interpretation, or substantive drafting. The authors take full responsibility for all content and have verified all facts and citations.

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

# Appendix

This supplementary document provides a backbone comparison (Appendix A), implementation and prompt details (Appendix B, C), and detailed feature analyses (Appendix D, E). We also report computational costs (Appendix F) and clarify feature roles (Appendix G). Finally, we present additional experiments on scalability and discuss limitations in Appendix I.

## A ADDITIONAL ANALYSIS OF OTHER BACKBONES

Table A: Comparison of large-scale backbones. "*" denotes U-Net only. G: generative, D: discriminative.

| Model | Params | Train data | Type |
|---|---|---|---|
| Ours (SD-v1.5*) | ≈0.8B | LAION-2B | G |
| DINOv2-L/14 | ≈0.3B | LVD-142M | D |
| DINOv2-G/14 | ≈1.1B | LVD-142M | D |
| OpenCLIP-L/14 | ≈0.4B | LAION-2B | D |
| OpenCLIP-H/14 | ≈1.0B | LAION-2B | D |

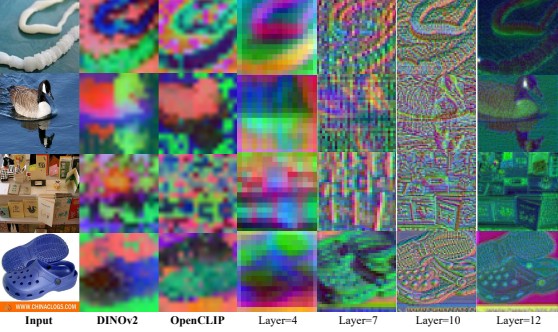

Figure A: PCA feature visualization of various backbones.

**Examining large-scale backbones.** While Fig. 2 in the main paper highlights performance gaps across large-scale backbones, Tab. A provides explicit differences in parameter scale, training data, and training objectives for a clearer understanding of our comparisons. In particular, DINOv2 (Oquab et al., 2023) is pre-trained on LVD-142M, a corpus curated using benchmark-relevant datasets (*e.g.,* ImageNet-1K/22K, CUB-200) as retrieval anchors. Although not directly trained on these benchmarks, this retrieval-based curation strategy injects domain-aligned images, thereby improving recognition performance on these target sets (Rodríguez-de Vera et al., 2025; Abbas et al., 2024). Combined with its discriminative self-supervised objective, this ensures that discriminative signals tailored to the specific benchmarks are inherently embedded in the model, resulting in superior initial accuracy as shown in Fig. 2 (left). In contrast, SD is trained on LAION-2B (Schuhmann et al., 2022) with a generative denoising objective, focusing on reconstructing images conditioned on text rather than directly optimizing for classification (Mukhopadhyay et al., 2024; Zhang et al., 2024b). Owing to this property, SD enables both inversion and generation processes to produce meaningful multi-scale representations within a unified backbone, without requiring extra modules. As evidenced by the pilot study (Fig. 2 in the main paper), SD-derived representations preserve knowledge more effectively and exhibit greater flexibility than discriminative counterparts. Moreover, when fully leveraged through our proposed framework, these representations ultimately surpass even larger discriminative backbones such as DINOv2-G/14 in later sessions.

**Qualitative comparison.** The qualitative PCA analysis shown in Fig. A underscores the advantages of extracting multi-scale representations from SD. By utilizing multiple layers within SD, we capture diverse visual characteristics—from global structural patterns in earlier layers to detailed, fine-grained cues in later layers. These multi-scale representations yield semantically rich features, which our proposed model effectively leverages to support improved FSCIL performance.

**Backbone comparison details.** For the backbone comparison in the main Fig. 2, we use Stable Diffusion v1.5 (Rombach et al., 2022) as both the baseline and our method's backbone. The discriminative baselines employ DINOv2 (Oquab et al., 2023) and OpenCLIP (Ilharco et al., 2021) with ViT-L/14 architectures, supplemented by larger variants (DINOv2-G/14 and OpenCLIP-H/14) to examine scaling effects (Tab. A). Concretely, we loaded the official pre-trained checkpoints and kept all backbones frozen during training, updating only our additional networks with the Dot-Regression (DR) loss. All backbones use same lightweight modules, including an aggregation module, a convolutional layer, and an MLP layer, to ensure a fair comparison. When using ViT backbones, we further exploit intermediate transformer blocks to construct multi-level representations that mimic the multi-scale features used in SD, while avoiding any backbone-specific fine-tuning or extra techniques that might bias the comparison.

Table B: **FSCIL results on CIFAR-100**. AA denotes the average accuracy, and FI represents the accuracy improvement at the final session.[†]results from Yang et al. (2023). **Bold**=best;underline=second best per column.

| Methods | Session Acc. (%) | | | | | | | | | AA (%) | FI |
|---|---|---|---|---|---|---|---|---|---|---|---|
| | 0 | 1 | 2 | 3 | 4 | 5 | 6 | 7 | 8 | | |
| TOPIC[†] (Tao et al., 2020) | 64.1 | 55.9 | 47.1 | 45.2 | 40.1 | 36.4 | 34.0 | 31.6 | 29.4 | 42.6 | +31.2 |
| CEC[†] (Zhang et al., 2021) | 73.1 | 68.9 | 65.3 | 61.2 | 58.1 | 55.6 | 53.2 | 51.3 | 49.1 | 59.5 | +11.5 |
| FACT (Zhou et al., 2022) | 74.6 | 72.1 | 67.6 | 63.5 | 61.4 | 58.4 | 56.3 | 54.2 | 52.1 | 62.2 | +8.5 |
| ERDFR (Liu et al., 2022) | 74.4 | 70.2 | 66.5 | 62.5 | 59.7 | 56.6 | 54.5 | 52.4 | 50.1 | 60.8 | +10.5 |
| ALICE[†] (Peng et al., 2022) | 79.0 | 70.5 | 67.1 | 63.4 | 61.2 | 59.2 | 58.1 | 56.3 | 54.1 | 63.2 | +6.5 |
| NC-FSCIL[†] (Yang et al., 2023) | 82.5 | 76.8 | 73.3 | 69.7 | 66.2 | 62.9 | 61.0 | 59.0 | 56.1 | 67.5 | +4.5 |
| BiDistFSCIL (Zhao et al., 2023) | 79.5 | 75.4 | 71.8 | 68.0 | 65.0 | 62.0 | 60.2 | 57.7 | 55.9 | 66.2 | +4.7 |
| SAVC (Song et al., 2023) | 78.8 | 73.3 | 69.3 | 64.9 | 61.7 | 59.3 | 57.1 | 55.2 | 53.1 | 63.6 | +7.5 |
| OrCo (Ahmed et al., 2024) | 80.1 | 68.2 | 67.0 | 61.0 | 59.8 | 58.6 | 57.0 | 55.1 | 52.2 | 62.1 | +8.4 |
| CLOSER (Oh et al., 2024) | 75.7 | 71.8 | 68.3 | 64.6 | 61.9 | 59.3 | 57.5 | 55.4 | 53.3 | 63.1 | +7.3 |
| Yourself (Tang et al., 2024) | 82.9 | 76.3 | 72.9 | 67.8 | 65.2 | 62.0 | 60.7 | 58.8 | 56.6 | 67.0 | +4.0 |
| Tri-WE (Lee et al., 2025) | 81.9 | 77.6 | 74.5 | 71.1 | 66.8 | 64.0 | 62.1 | 61.7 | 58.2 | 68.7 | +2.4 |
| **Ours** | **83.1** | 75.4 | 70.9 | 68.4 | 66.8 | 65.3 | 64.9 | 62.7 | 60.6 | 68.7 | – |

# B  ADDITIONAL IMPLEMENTATION DETAILS

**Implementation details.** We provide implementation details for our main experiments in Sec. 5 and CIFAR-100 results (Tab. B). We use AdamW (Loshchilov & Hutter, 2019) with weight decay $10^{-4}$ and initial learning rates of $3 \times 10^{-3}$ for MLP layer ($g^{\text{MLP}}$) and $1 \times 10^{-3}$ for the aggregation network ($g^{\text{agg}}$). During base session ($\mathcal{S}^0$) training, we train all components ($g^{\text{agg}}$, $g^{\text{conv}}$, $g^{\text{MLP}}$) using primarily $\mathbf{F}_{\text{inv}}$ and $\mathbf{F}_{\text{syn}}$ using different epoch numbers: 70 epochs (for CUB-200), 200 epochs (for miniImageNet), and 300 epochs (for CIFAR-100), respectively. For miniImageNet and CIFAR-100, we additionally incorporate $\mathbf{F}_{\text{aug}}$ during the final training phase (approximately 20% of the base training duration) to enhance the generalization ability. For incremental sessions ($\mathcal{S}^{i \geq 1}$), only MLP layer is trainable while $g^{\text{agg}}$ and $g^{\text{conv}}$ remain frozen. We apply standard data augmentation techniques including random resizing, rotation, color jittering, and horizontal flipping following (Yang et al., 2023) excluding incremental sessions.

**Architectural details.** Detailed architectures of our subnetworks (*i.e.,* $g^{\text{conv}}$, $g^{\text{MLP}}$, and $g^{\text{agg}}$) are visualized in Fig. B. The convolutional layer ($g^{\text{conv}}$) consists of a convolutional layer followed by Batch Normalization (BN), SiLU activation, and a bottleneck layer for efficient feature projection. The MLP layer ($g^{\text{MLP}}$) is structured with two linear layers, each followed by Group Normalization (GN) and SiLU activation, incorporating a residual connection for enhanced gradient flow. The aggregation network ($g^{\text{agg}}$), inspired by prior works (Luo et al., 2024; Wang et al., 2024), integrates multi-layer SD features through layer-wise upsampling, convolutional processing, element-wise addition, and final L2 normalization. Our complete processing pipeline follows the sequence: aggregation network $\rightarrow$ convolutional layer $\rightarrow$ MLP layer with one residual connection, creating a lightweight architecture with approximately 6M trainable parameters as shown in the main Fig. 3.

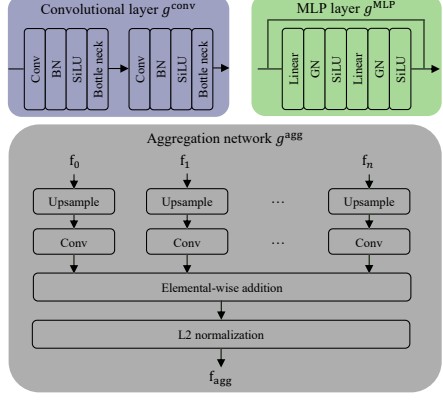

Figure B: **Architectural details of our method.** Except for the backbone, our method consists of mainly three components: (1) convolutional layer ($g^{\text{conv}}$), (2) MLP layer ($g^{\text{MLP}}$), and (3) aggregation network ($g^{\text{agg}}$).

**Class-specific prompt details.** For each class $c$, we learn a class-specific embedding $\mathbf{w}_c^*$ and its corresponding prompt $\mathbf{p}_c^*$ via textual inversion (Gal et al., 2022). Since $\mathbf{w}_c^*$ is optimized to capture fine-grained class semantics starting from its initialization, the initializer should provide a clear and unambiguous semantic anchor for stable optimization. In our benchmarks, class labels are often written using multiple words or synonym phrases (*e.g.,* "ladybug, ladybeetle, lady beetle, ladybird, ladybird beetle" or "Walker

Table C: Hyper-parameters for optimizing class-specific prompts.

| Name | CUB-200 | miniImageNet | CIFAR-100 |
|---|---|---|---|
| Batch size | 1 | 1 | 1 |
| Warm-up iter. | 200 | 200 | 200 |
| Learning rate | $10^{-4}$ | $10^{-4}$ | $10^{-4}$ |
| Training iter. | 2000 | 2000 | 2000 |
| Resolution | 512 | 512 | 384 |
| Emb. vec. size | 5 | 7 | 7 |

---

**Algorithm 1:** Overall training procedure

---

```
# Datasets: Base D^0, Incremental {D^1, ..., D^i}
# Networks: g^agg, g^conv, g^MLP
# SD: Frozen Stable Diffusion backbone

freeze(SD)  # Always frozen
for session, data in enumerate([D^0]+[D^1, ..., D^i]):
    # Base session S^0
    if session == 0:
        # Optimize class-specific prompts p* for base data
        learn_prompt(data)

        # Train g^agg, g^conv, g^MLP
        train(F_inv, F_syn, F_aug)

        # Freeze for incremental sessions
        freeze(g^agg, g^conv)

    # Incremental sessions S^{i>=1}
    else:
        # Freeze previous g^MLP to serve as a teacher
        g_teacher = copy_and_freeze(g^MLP)

        # Optimize class-specific prompts p* for few-shot data
        learn_prompt(data)

        # Train g^MLP network (F_gen is updated with knowledge distillation from g_teacher)
        train(F_inv, F_syn, F_gen, F_aug)
```

---

hound, Walker foxhound"), which may introduce semantic ambiguity. To address this issue, we utilize a large language model (OpenAI, 2023) to manually consolidate each label into a single word (*e.g.*, "Walker hound, Walker foxhound" → "dog"). Each single word is initially assigned for $\mathbf{w}_c^*$, which is subsequently optimized using diverse text templates (*e.g.*, "a photo of a {}", "a rendering of a {}") to produce the final class-specific prompt $\mathbf{p}_c^*$. The complete set of templates and corresponding hyperparameters used for this optimization are provided in Tab. C and E, respectively.

**Overall training procedure.** We summarize the overall training procedure in Algorithm 1, which outlines the workflow across all sessions. In the base session ($\mathcal{S}^0$), we first optimize class-specific prompts $\mathbf{p}^*$ using real images. We then train lightweight networks $g^{\text{agg}}$, $g^{\text{conv}}$, and $g^{\text{MLP}}$ using inversion ($\mathbf{F}_{\text{inv}}$), synthetic ($\mathbf{F}_{\text{syn}}$), and augmented ($\mathbf{F}_{\text{aug}}$) features, while the SD backbone remains frozen. After completing the base-session training, $g^{\text{agg}}$ and $g^{\text{conv}}$ are frozen. In the incremental sessions ($\mathcal{S}^{i \geq 1}$), we optimize class-specific prompts $\mathbf{p}^*$ for few-shot real images with new classes. The model is then trained by updating only $g^{\text{MLP}}$, using all types of features ($\mathbf{F}_{\text{inv}}, \mathbf{F}_{\text{syn}}, \mathbf{F}_{\text{aug}}, \mathbf{F}_{\text{gen}}$). Here, $\mathbf{F}_{\text{gen}}$ is extracted through the pure-noise generation process ($\hat{z}_{\text{T}} \rightarrow \hat{z}_0$) conditioned on the class-specific prompts $\mathbf{p}^*$ obtained for previous sessions. For training, $\mathbf{F}_{\text{inv}}$, $\mathbf{F}_{\text{syn}}$, and $\mathbf{F}_{\text{aug}}$ use the DR loss (Yang et al., 2023), while $\mathbf{F}_{\text{gen}}$ use a knowledge distillation loss with the previous session's frozen $g^{\text{MLP}}$ as the teacher.

**Multi-scale feature layer selection.** We extract multi-scale features from SD's U-Net via a single-step inversion using a null text prompt. As mentioned in the main paper (Sec. 4.1), we exploit intermediate representations from U-Net layers 4–12, which strike a balance between detail and abstraction, while excluding the lowest-resolution layers 1–3. Ablation studies on CUB-200 confirm that this layer selection yields the best base-session accuracy (Tab. D).

Table D: CUB-200 base session performance. Layers denote SD U-Net ranges.

| Layers | Accuracy (%) |
|---|---|
| Full (1–12) | 80.69 |
| **4–12 layers** | **82.06** |
| 7–12 layers | 81.08 |
| 10–12 layers | 73.25 |

## C EFFECTIVENESS OF THE OPTIMIZED PROMPT

Here, we highlight the importance of utilizing optimized class-specific prompts $\mathbf{p}^*$ through qualitative comparisons. We visually compare two prompt strategies: features extracted using simple text prompts (*e.g.*, "A photo of {class-name}") versus features extracted using our optimized, class-specific prompts $\mathbf{p}^*$ (described in the main Sec. 4.2). As illustrated in Fig. D and Fig. E, when

conditioned on naïve prompts, SD frequently generates images that either roughly capture overall appearance but fail to generate precise object attributes (columns 2), or completely fail to reproduce objects related to the given prompt (columns 5). In contrast, employing optimized class-specific prompts $\mathbf{p}^*$ significantly enhances generative quality, enabling precise denoising of latent variables from $\mathbf{z}_T$ to $\mathbf{z}_0$. Importantly, these generated images are shown purely for visualization purposes to demonstrate prompt effectiveness—our method extracts multi-scale features directly from the denoising process without requiring actual image synthesis, thus eliminating storage overhead while preserving the semantic richness of the generative features.

## D    COMPARISON BETWEEN DIFFUSION FEATURES

Table E: **Examples of template for textual inversion**. "{}" is a placeholder for the class per session.

| Templates |
|---|
| *"a photo of a {}", "a rendering of a {}", "a cropped photo of the {}", "the photo of a {}", "a photo of a clean {}", "a photo of a dirty {}", "a dark photo of the {}", "a photo of my {}", "a photo of the cool {}", "a close-up photo of a {}", "a bright photo of the {}", "a cropped photo of a {}", "a photo of the {}", "a good photo of the {}", "a photo of one {}", "a close-up photo of the {}", "a rendition of the {}", "a photo of the clean {}", "a rendition of a {}", "a photo of a nice {}", "a good photo of a {}", "a photo of the nice {}", "a photo of the small {}", "a photo of the weird {}", "a photo of the large {}", "a photo of a cool {}", "a photo of a small {}"* |

Table F: Approximate number of trainable parameters of representative FSCIL methods on CUB-200.

| Methods | Venue | Trainable params. |
|---|---|---|
| FACT | CVPR'22 | $\approx$ 12M |
| ALICE | ECCV'22 | $\approx$ 42M |
| NC-FSCIL | ICLR'23 | $\approx$ 16M |
| SAVC | CVPR'23 | $\approx$ 24M |
| OrCo | CVPR'24 | $\approx$ 13M |
| CLOSER | ECCV'24 | $\approx$ 12M |
| Yourself | ECCV'24 | $\approx$ 12M |
| Ours | – | $\approx$ 6M |

**Visualization of multi-scale features.** We present detailed visualizations of the multi-scale SD features utilized in our framework in Fig. F. We visualize the four feature types (*i.e.,*, $\mathbf{F}_{inv}$, $\mathbf{F}_{syn}$, $\mathbf{F}_{aug}$, and $\mathbf{F}_{gen}$) extracted from different layers (4 to 12) of SD, given the same input image on CUB-200. For the augmented feature $\mathbf{F}_{aug}$, we inject Gaussian noise at timestep 0.5T (halfway point of the diffusion process) to clearly illustrate how partial noise affects the resulting feature representation. The generated images shown in Fig. F result from the generative feature extraction process ($\mathbf{F}_{gen}$), obtained using optimized class-specific prompts $\mathbf{p}^*$ for Herring_Gull and Ring_billed_Gull. Importantly, these generated images serve solely for visualization purposes and are not utilized during training or inference—our method operates directly on the extracted features without requiring image synthesis.

**Visualization across noise injection timesteps.** Additionally, we visually show how injected noise at intermediate timesteps $t$ over the full diffusion time T impacts the augmented feature $\mathbf{F}_{aug}$. As shown in Fig. G, we illustrate $\mathbf{F}_{aug}$ at four distinct timesteps: $t = \{\frac{T}{4}, \frac{2T}{4}, \frac{3T}{4}, T\}$. The augmentation process begins by injecting Gaussian noise into the original latent at timestep $t$, then applies partial denoising back to $t = 0$ to extract features from layers $\mathbf{f}'_4$ to $\mathbf{f}'_{12}$. For each timestep, we provide the original images and their corresponding denoised outputs (*i.e.,*, generated image; for visualization purposes only). The augmented feature $\mathbf{F}_{aug}$ are presented from top-left to bottom-right, corresponding sequentially to the extraction layers. Importantly, our method uses only these extracted features during training—the generated images serve solely to demonstrate the feature extraction quality at different noise levels.

## E    ADDITIONAL DISCUSSION OF ONE-STEP FEATURES

In Sec. 4.1 of the main paper, we motivated the use of one-step diffusion features for both $\mathbf{F}_{inv}$ and $\mathbf{F}_{syn}$ during training. Here, we provide further evidence through simple latent-space exploration. As shown in Fig. C, latents at small timesteps (*e.g.,* $t{=}0, 5$) retain clear semantic structure, while those at larger timesteps (*e.g.,* $t{=}20, 50$) become progressively dominated by noise and lose semantic meaning. For instance, Ring_billed_Gull and Black_footed_Albatross remain semantically recognizable at $t{=}0$ and $t{=}5$, but lose class-relevant



Figure C: Visualization of latent variable across different diffusion timesteps $t$.

patterns as noise increases, and `Red_breasted_Merganser` becomes indistinguishable by $t = 50$. This observation supports our choice to consistently adopt the minimal timestep representation.

## F    TRAINABLE PARAMETERES AND COMPUTATIONAL COSTS

**Trainable parameters.**    Unlike most FSCIL frameworks that fine-tune an entire network, our method freezes the text-to-image diffusion backbone and updates only an aggregation network and a lightweight neck (Conv and MLP layers). To quantify this design's impact, Tab. F compares the number of trainable parameters across representative methods, including FACT (Zhou et al., 2022), ALICE (Peng et al., 2022), NC-FSCIL (Yang et al., 2023), SAVC (Song et al., 2023), OrCo (Ahmed et al., 2024), CLOSER (Oh et al., 2024), and Yourself (Tang et al., 2024). Most existing approaches require at least 12M parameters since they fine-tune a ResNet-series backbone end-to-end. In contrast, our method keeps the diffusion backbone frozen and optimizes only the aggregation networks and neck, achieving the smallest footprint ($\approx$6M). This demonstrates that our design ensures parameter efficiency.

**Inference costs.** We compute the inference computational costs (in GFLOPs) against baselines including OpenCLIP (Ilharco et al., 2021) and DINOv2 (Oquab et al., 2023). During training, our method extracts four feature types ($\mathbf{F}_{inv}$, $\mathbf{F}_{syn}$, $\mathbf{F}_{gen}$, $\mathbf{F}_{aug}$) through multiple forward passes of the SD backbone; however, at inference, the model uses only a single VAE encoding followed by one U-Net forward pass to compute $\mathbf{F}_{inv}$. In terms of inference complexity, OpenCLIP, DINOv2, and our method require approximately 85, 180, and 890 GFLOPs per image, respectively. Our method incurs higher computational costs due to the frozen SD backbone that we leverage for extracting multi-scale generative representations, while our trainable parameters remain minimal ($\approx$6M). This computational overhead remains a known challenge in diffusion-based downstream tasks, motivating future research on efficiency.

**Base session costs.** The base session fine-tuning on CUB-200 requires approximately 3 hours using 4 A100 GPUs. This efficiency is achieved by freezing the heavy Stable Diffusion backbone and optimizing only the lightweight trainable modules ($\approx$6M parameters).

## G    DISTINCTIVE ROLES OF EACH FEATURE TYPE

Our framework extracts four feature types from Stable Diffusion, each serving a distinct role in addressing FSCIL challenges: **Inversion feature ($\mathbf{F}_{inv}$)** captures image-conditioned structural representations through the forward diffusion process with null prompts. This feature preserves fine-grained visual details from real samples and serves as the primary representation during inference, providing stable knowledge across all sessions. **Synthetic feature ($\mathbf{F}_{syn}$)** introduces class-aware semantic information by conditioning on class-name prompts during one-step generation. By leveraging SD's text-conditioned prior, this feature enriches the representation with semantic diversity while maintaining alignment with the inversion pathway. **Generative feature ($\mathbf{F}_{gen}$)** enables latent-space replay without storing synthetic images. Extracted from full generation processes conditioned on class-specific prompts $\mathbf{p}^*$, this feature replays learned class concepts for effective forgetting mitigation in incremental sessions. **Augmented feature ($\mathbf{F}_{aug}$)** applies controlled noise injection and partial denoising with class-specific prompts $\mathbf{p}^*$ to introduce semantic variations while preserving structural fidelity. This feature enhances generalization under limited few-shot samples by balancing fidelity and diversity through multi-interval noise scheduling.

## H    ADDITION EXPERIMENTS USING LARGER DATASET AND DIFFERENT BACKBONE

**Scalability to large-scale benchmark.** To examine the scalability of our framework beyond standard FSCIL datasets, we further evaluate it on ImageNet-100 following the protocol of Zhao et al. (2024). ImageNet-100 provides a substantially more challenging setting, containing a considerably larger training corpus (approximately 1,300 images per class for the base session) with greater visual diversity. In this setting, our method maintains strong performance across sessions, as shown in Tab. G. Specifically, we achieve 74.58% final accuracy and 81.95% average accuracy (AA.), clearly outperforming the recent state-of-the-art LRT (Zhao et al., 2024) and other competitive

Table G: **FSCIL results on ImageNet-100**. Top-1 accuracies (%) for sessions 0–8 (0 = base session). AA. denotes average accuracy across all sessions. FI calculates the improvement of our method in the last session compared to previous methods. [†]results from Zhao et al. (2024). **Bold**=best; underline=second best per column.

| Methods | Session Acc. (%) | | | | | | | | | AA.(%) | FI |
|---|---|---|---|---|---|---|---|---|---|---|---|
| | 0 | 1 | 2 | 3 | 4 | 5 | 6 | 7 | 8 | | |
| CEC[†] (Zhang et al., 2021) | 84.77 | 80.03 | 76.66 | 73.10 | 69.30 | 65.88 | 64.27 | 62.91 | 60.04 | 70.77 | +14.54 |
| FACT[†] (Zhou et al., 2022) | 86.00 | 80.94 | 77.66 | 75.34 | 70.40 | 66.72 | 64.82 | 63.15 | 60.98 | 71.67 | +13.60 |
| LRT[†] (Zhao et al., 2024) | 91.43 | 87.03 | 83.83 | 79.34 | 76.15 | 72.05 | 70.18 | 68.52 | 65.90 | 77.16 | +8.68 |
| **Ours** | **92.80** | **89.20** | **85.37** | **82.83** | **79.85** | **78.54** | **77.67** | **76.74** | **74.58** | **81.95** | – |

Table H: **FSCIL results using DiT backbone on CUB-200**. Top-1 accuracies (%) for sessions 0–10 (0 = base session). Comparison with NC-FSCIL (Yang et al., 2023) using a DiT-based diffusion transformer (Labs et al., 2025) as a frozen backbone. Only the core generative feature $F_{gen}$ is used for this evaluation. AA. denotes average accuracy. **Bold**=best per column.

| Method | Session Acc. (%) | | | | | | | | | | | AA.(%) |
|---|---|---|---|---|---|---|---|---|---|---|---|---|
| | 0 | 1 | 2 | 3 | 4 | 5 | 6 | 7 | 8 | 9 | 10 | |
| NC-FSCIL (Yang et al., 2023) | **75.50** | **61.22** | 54.81 | 51.66 | 49.85 | 46.21 | 44.04 | 42.56 | 39.32 | 38.51 | 37.08 | 49.16 |
| **Ours** | **75.50** | 61.12 | **57.23** | **52.31** | **51.07** | **47.23** | **45.02** | **42.84** | **40.34** | **40.32** | **38.71** | **50.15** |

methods (Zhang et al., 2021; Zhou et al., 2022). These results indicate that our framework scales effectively, sustaining both stability and plasticity even under large-scale dataset.

**Adaptability to different architectures.** To further examine whether our framework generalizes beyond Stable Diffusion, we additionally evaluate it using a DiT-based diffusion transformer (Labs et al., 2025), one of the recent large generative models, as the frozen backbone. Given the substantial architectural differences of DiT, we use only the generative feature $F_{gen}$ to verify whether the core part of our strategy remains effective under this backbone. As shown in Tab. H, our method achieves higher accuracy in most incremental sessions, indicating that the core idea of our framework generalizes reasonably well to such a different generative backbone. While a comprehensive adaptation for diffusion transformers would require deeper investigation of their representational features (Gan et al., 2025) and thus remains beyond the scope of this work, our results demonstrate promising potential for extending the framework to a broader family of generative backbones in future research.

# I   LIMITATIONS

Although our method shows strong performance, the reliance on a large diffusion backbone inevitably increases computational demands. While we presented a streamlined variant to mitigate this, further optimization and efficient techniques remain promising directions.

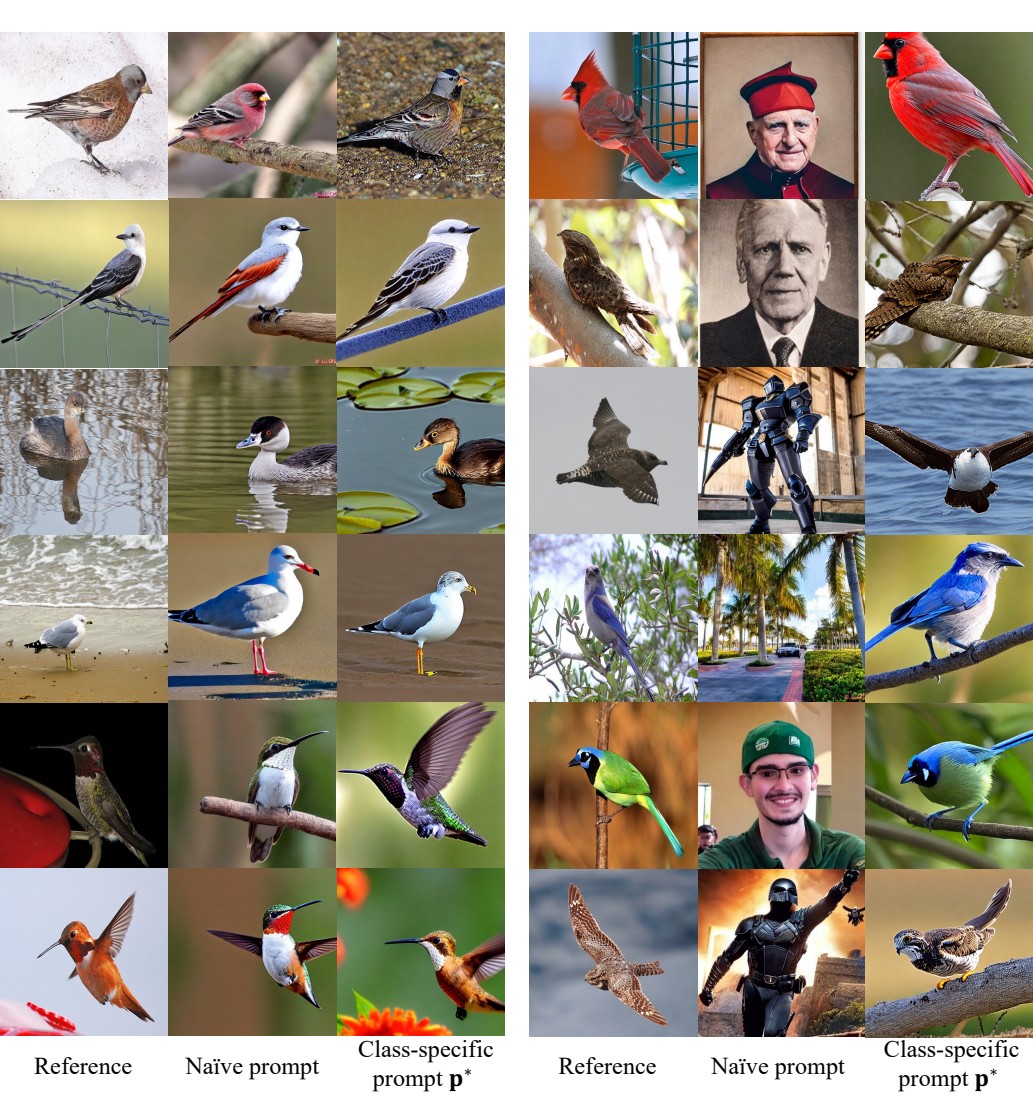

Reference    Naïve prompt    Class-specific prompt **p**$^*$    Reference    Naïve prompt    Class-specific prompt **p**$^*$

Figure D: **Visualization of synthesized images for generative features F$_{\text{gen}}$ on CUB-200** (Wah et al., 2011). From left to right, we show the reference images, generated images using a naïve prompt (*e.g.,* "A photo of {class-name}"), and generated images obtained using our optimized, class-specific prompt **p**$^*$. Columns 2 and 5 illustrate typical failures when employing naïve prompt, where synthesized images either lack precise details (column 2) or fail entirely to match the reference class (column 5). In contrast, Columns 3 and 6 clearly demonstrate improved synthesis quality when optimized class-specific prompt **p**$^*$ are utilized. These visualizations explicitly highlight the necessity of optimized prompt for accurate generative feature extraction. (*Note*: This figure is provided solely for qualitative visualization purposes.)

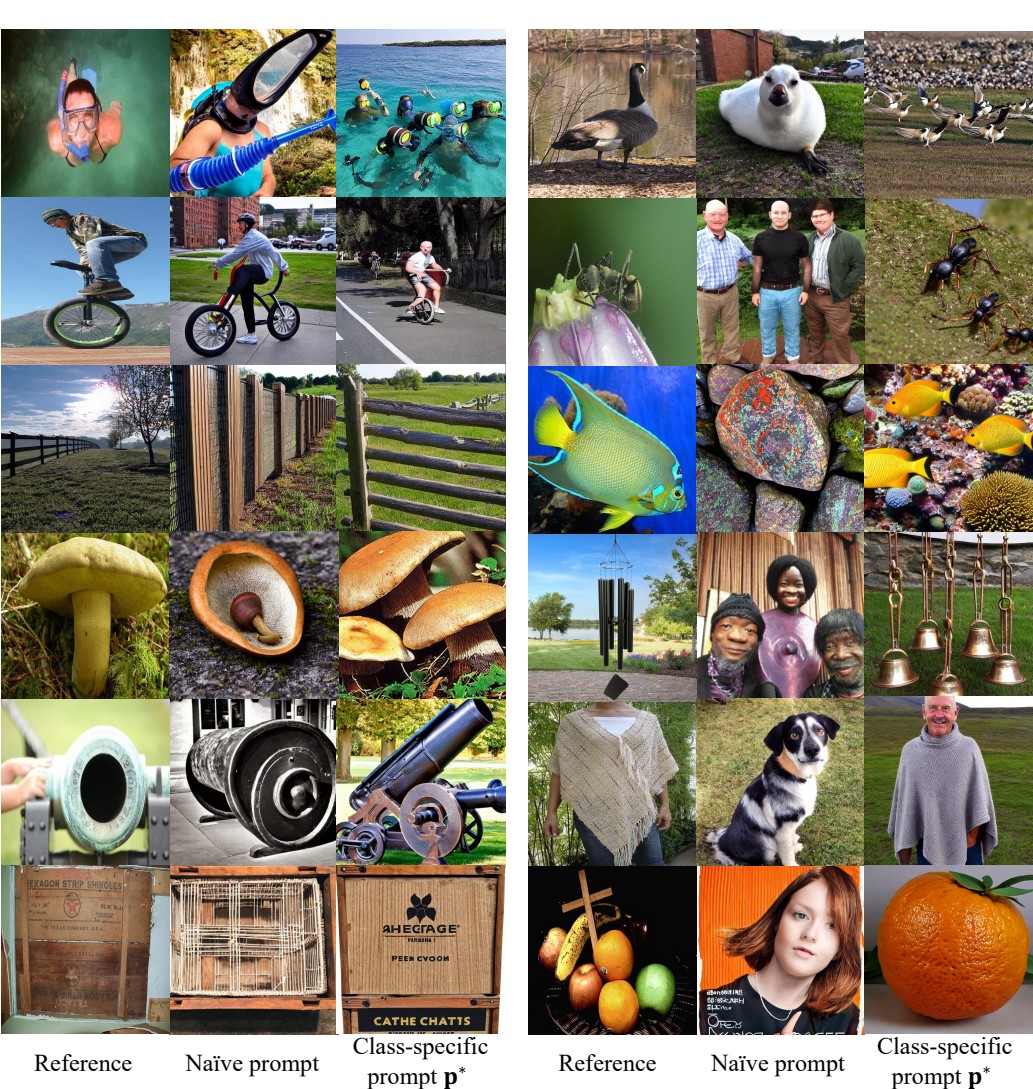

|  |  | Class-specific |  |  | Class-specific |
| Reference | Naïve prompt | prompt $\mathbf{p}^*$ | Reference | Naïve prompt | prompt $\mathbf{p}^*$ |

Figure E: **Visualization of synthesized images for generative features $\mathbf{F}_{\text{gen}}$ on miniImageNet** (Russakovsky et al., 2015b). From left to right, we show the reference images, images generated by naïve prompt, and images generated by optimized class-specific prompt $\mathbf{p}^*$. Naïve prompt-based generation often leads to insufficient detail capture (column 2) or completely incorrect generation (column 5). However, using optimized prompt $\mathbf{p}^*$ significantly improves the generative results, closely reflecting class-specific characteristics (column 3 and 6). We emphasize that these results are visualized solely to qualitatively illustrate differences in prompt effectiveness.

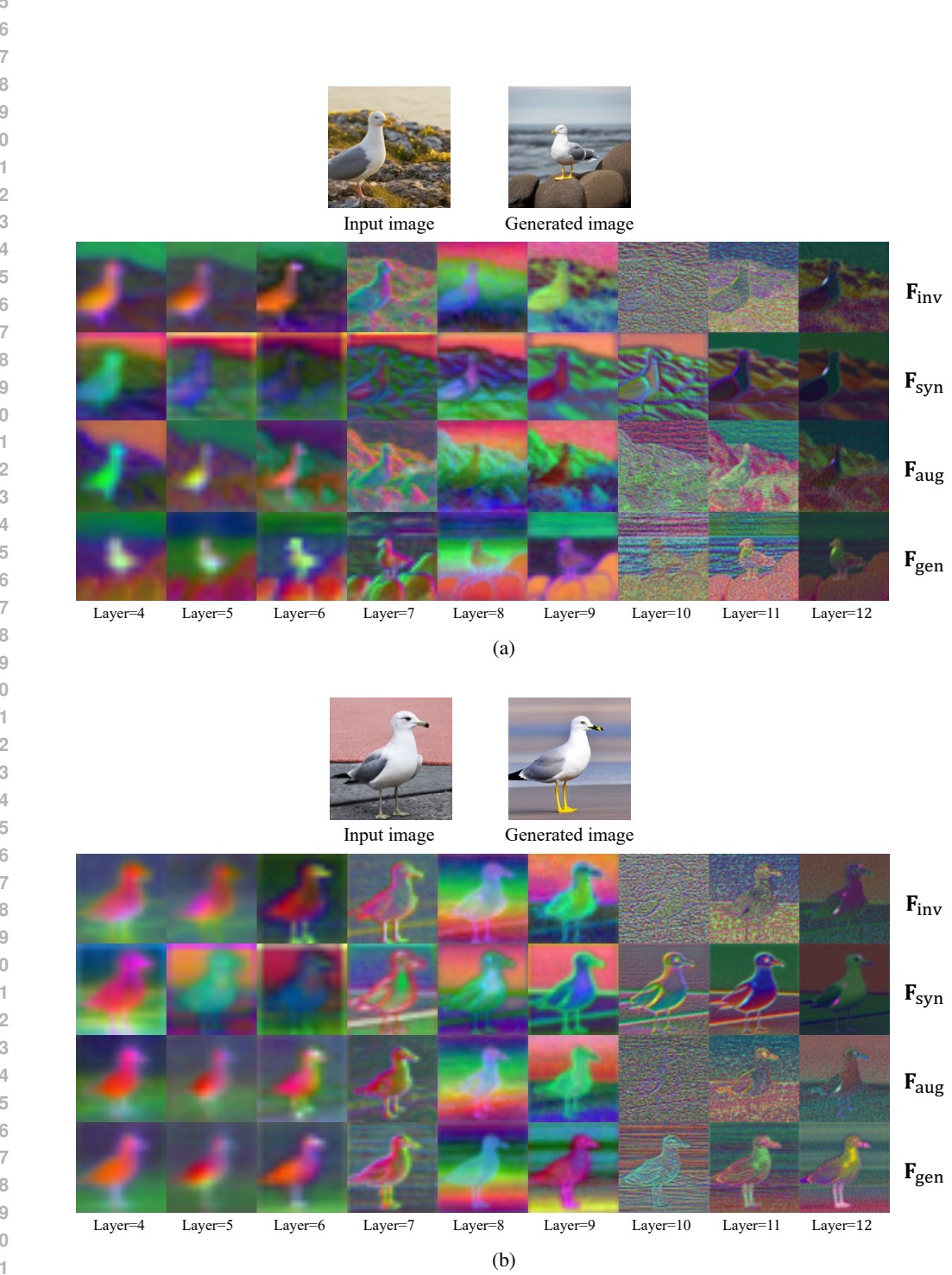

Figure F: **Example multi-scale features visualization for an image in CUB-200**: (a) `Herring_Gull` and (b) `Ring_billed_Gull`. Note that the generated images are shown for visualization purposes only.

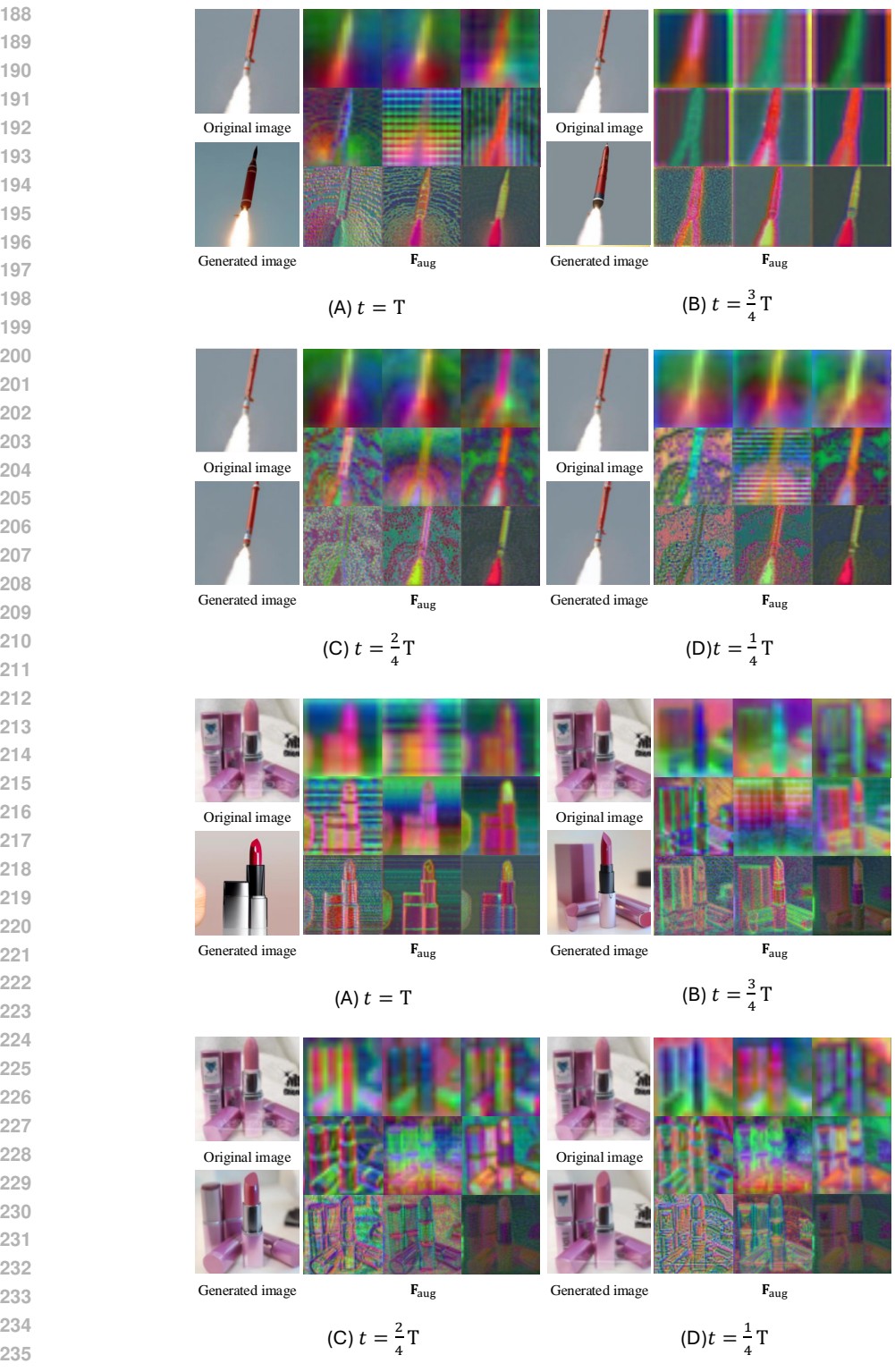

Figure G: SD intermediate feature visualization across different diffusion timesteps $t$. For each noise-injected timestep ($\frac{T}{4}, \frac{2T}{4}, \frac{3T}{4}, T$), we show original images and their corresponding denoised outputs (*i.e.,*, Generated image; for visualization purposes only). Visualizations of augmented features $\mathbf{F}_{aug}$ extracted from U-Net layers $\mathbf{f}'_4$ to $\mathbf{f}'_{12}$ during the partial denoising process are sequentially arranged from top-left to bottom-right. These images demonstrate feature extraction quality—our method operates directly on the features without requiring image synthesis.

