# OpenReview forum: "Beyond Synthetic Replays: Turning Diffusion Features into Few-Shot Class-Incremental Learning Knowledge"
_ICLR.cc/2026/Conference — Submitted to ICLR 2026_

### Official Review · Reviewer_7vfk · 2025-10-27

**Soundness:** 3
**Presentation:** 3
**Contribution:** 3
**Rating:** 4
**Confidence:** 3

**Summary:**

This paper introduces Diffusion-FSCIL, a novel few-shot class-incremental learning (FSCIL) framework that leverages Stable Diffusion (SD) as a unified frozen backbone for feature extraction. Unlike prior works that use SD solely for generating synthetic replay images, this approach extracts four complementary types of features (inversion, synthetic, augmented, and generative) directly from the diffusion process in latent space. The method achieves state-of-the-art performance on CUB-200, miniImageNet, and CIFAR-100, while maintaining efficiency with only ~6M trainable parameters.

**Strengths:**

1.The idea of using SD as a frozen feature extractor instead of a generative replay buffer is original and well-motivated.
2. The use of multi-scale U-Net features from SD is technically sound and well-justified. The class-specific prompt optimization and controlled noise injection strategies are clever and effective.
3. The paper is well-written, with clear structure and intuitive illustrations.

**Weaknesses:**

1. While the empirical results are strong, the paper lacks theoretical analysis on why  SD features are more robust to forgetting compared to discriminative models like DINOv2.
2. All experiments are conducted on small-scale image classification datasets.
3. There is no discussion on how the method would adapt to other generative models (e.g., DiT, CLIP-guided diffusion, autoregressive models).

**Questions:**

1. Why does Stable Diffusion (SD) retain knowledge better than discriminative models like DINOv2 or CLIP in incremental sessions?
2. Why use four types of features instead of learning a single unified representation?

---

> ### Author Response · Authors · 2025-11-22
>
> > W1) While the empirical results are strong, the paper lacks theoretical analysis on why SD features are more robust to forgetting compared to discriminative models like DINOv2. \
> > Q1) Why does Stable Diffusion (SD) retain knowledge better than discriminative models like DINOv2 or CLIP in incremental sessions?
>
> `Answer:` Thanks for the constructive comment.
>
> Discriminative models such as DINOv2 and CLIP are forward-only encoders that produce sharply class-aligned decision features [1], learning representations optimized for strong class separation. These properties help them react strongly to a few new examples, but without any auxiliary mechanism to prevent forgetting, incremental updates tend to aggresively overwrite older classes, resulting in faster forgetting.
>
> Another aspect is that conventional generative replay exhibits representational discrepancies because it relies on separate discriminative and generative backbones. In contrast, we propose a unified backbone, where augmented features are generated by fully exploiting these dual pathways. Since all of features originate from the same model, the resultant updates are less aggressive and better aligned with previously learned representations, helping the model retain earlier classes even when new classes arrive with limited data (e.g., FSCIL situation). We think that this is also demonstrated in Figure 2 or Figure 5 that shows that naïve application of generative replay scheme using SD features does not well operate; while our method based on unified backbones with 4 types features works effectively.
>
> **References**:
> [1] Zhang et al., A Tale of Two Features: Stable Diffusion Complements DINO for Zero-Shot Semantic Correspondence, NeurIPS 2023.
>
> ---
>
> > W2) All experiments are conducted on small-scale image classification datasets.
>
> `Answer:` Thank you for pointing this out. We used the standard FSCIL benchmarks commonly adopted in recent works such as [1, 2]. We additionally evaluated our method on ImageNet100 following the FSCIL protocol used in LRT [3]. In this setting, LRT reports 65.90% final accuracy and 77.16% average accuracy, while ours achieves 74.58% final accuracy and 81.95% average accuracy. The per-session accuracies are:
>
> | Method | S0    | S1    | S2    | S3    | S4    | S5    | S6    | S7    | S8    | Avg.  |
> |--------|------:|------:|------:|------:|------:|------:|------:|------:|------:|------:|
> | LRT [3]  | 91.43 | 87.03 | 83.83 | 79.34 | 76.15 | 72.05 | 70.18 | 68.52 | 65.90 | 77.16 |
> | Ours     | **92.80** | **89.20** | **85.37** | **82.83** | **79.85** | **78.54** | **77.67** | **76.74** | **74.58** | **81.95** |
>
> This result indicates that our approach also achieves strong performance on ImageNet100 under the large-scale FSCIL setting.
>
> **References:** \
> [1] Yang et al., “Neural Collapse Inspired Feature-Classifier Alignment for Few-Shot Class-Incremental Learning,” ICLR, 2023.  \
> [2] Tang et al., “Rethinking Few-Shot Class-Incremental Learning: Learning from Yourself,” ECCV, 2024.  \
> [3] Zhao et al., “Language-Inspired Relation Transfer for Few-Shot Class-Incremental Learning,” IEEE TPAMI, 2024.

---

> ### Author Response · Authors · 2025-11-22
>
> > W3) There is no discussion on how the method would adapt to other generative models (e.g., DiT, CLIP-guided diffusion, autoregressive models).
>
> `Answer 1:` We believe the proposed method can also be applied to other generative models that are sufficiently pre-trained on diverse datasets and learn semantic representations through their denoising-based generative objective. Diffusion models trained in this manner are known to encode high-level semantic cues [1] in their latent features, and Gan et al. [2] showed that DiT architectures also exhibit such diffusion-based semantics, suggesting that our approach may extend to a broader class of diffusion models beyond SD. Following the reviewer’s comment, we are additionally running an experiment with a DiT-based backbone and will include the results within the rebuttal period.
>
> **References:** \
> [1] Luo et al., “Diffusion Hyperfeatures: Searching Through Time and Space for Semantic Correspondence,” NeurIPS, 2023. \
> [2] Gan et al., “Unleashing Diffusion Transformers for Visual Correspondence by Modulating Massive Activations.”, NeurIPS 2025.
>
> ---
>
> > Q2) Why use four types of features instead of learning a single unified representation?
>
> `Answer:` We clarify why we maintain four distinct feature types rather than learning a single unified representation. Our unified SD framework leverages both inversion and generation pathways, and each feature type fulfills a different role required in FSCIL:
>
> - $\textbf{F}\_\text{inv}$: captures stable structural information from real images, providing the basic visual representation shared across sessions.
>
> - $\textbf{F}\_\text{syn}$: captures class-relevant semantics by providing the class name to the SD text encoder, allowing the representation to reflect SD’s semantic prior for that class.
>
> - $\textbf{F}\_\text{gen}$: captures past class concepts for latent replay, helping prevent forgetting without storing synthetic images.
>
> - $\textbf{F}\_\text{aug}$: introduces controlled variations of real samples to improve robustness, enabling better generalization from limited few-shot data.
>
> These feature types serve distinctive roles that cannot be fulfilled by a single unified representation. As shown in Tab. 3, removing any component noticeably degrades performance, confirming that all four are necessary for strong FSCIL performance. We added a summary of the roles of each feature type in Appendix G (L948–960).

---

> ### Comment · Reviewer_7vfk · 2025-11-28
>
> 1. you cite some papers, but the results are not in your papers. So are you going to supplement these experiments?

---

> ### Author Response · Authors · 2025-12-01
> **Answer & Additional DiT experiment remind**
>
> > you cite some papers, but the results are not in your papers. So are you going to supplement these experiments?
>
> `Answer`: Yes. We have added extended experiments to address the points raised in W2 and W3. Both experiments are included in the revised manuscript:
>
> - ImageNet-100 evaluation (W2): L966–993 and Tab. G
> - DiT backbone evaluation (W3): L994–1004 and Tab. H
>
> ---
> > W3) There is no discussion on how the method would adapt to other generative models (e.g., DiT, CLIP-guided diffusion, autoregressive models).
>
> `Answer 2:` We have completed the DiT backbone evaluation as mentioned in our previous response. When both methods are evaluated using the FLUX.1-based diffusion transformer [1] as the frozen backbone, our method consistently outperforms NC-FSCIL [2] across most incremental sessions:
>
> | Method   |   S0   |   S1  |   S2  |   S3  |   S4  |   S5  |   S6  |   S7  |   S8  |   S9  |  S10  |  Avg. |
> |----------|-------:|------:|------:|------:|------:|------:|------:|------:|------:|------:|------:|------:|
> | NC-FSCIL | **75.50**  | **61.22** | 54.81 | 51.66 | 49.85 | 46.21 | 44.04 | 42.56 | 39.32 | 38.51 | 37.08 | 49.16 |
> | Ours     | **75.50**  | 61.12 | **57.23** | **52.31** | **51.07** | **47.23** | **45.02** | **42.84** | **40.34** | **40.32** | **38.71** | **50.15** |
>
> Given the substantial architectural differences of DiT, we leverage $\textbf{F}_\text{gen}$, one of our core feature types. While full adaptation would require deeper investigation of DiT's representational features [3] and remains beyond this work's scope, these results demonstrate promising potential for extending our framework to a broader family of generative architectures. This evaluation was added to the revised manuscript at L994–1004 and Tab. H.
>
> **References:** \
> [1] Black Forest Labs, "FLUX.1 [dev]," https://huggingface.co/black-forest-labs/FLUX.1-dev, 2024. \
> [2] Yang et al., "Neural Collapse Inspired Feature-Classifier Alignment for Few-Shot Class-Incremental Learning," ICLR, 2023. \
> [3] Gan et al., "Unleashing Diffusion Transformers for Visual Correspondence by Modulating Massive Activations," NeurIPS, 2025.

---

### Official Review · Reviewer_SWdq · 2025-10-30

**Soundness:** 2
**Presentation:** 2
**Contribution:** 2
**Rating:** 4
**Confidence:** 4

**Summary:**

The paper proposes Diffusion-FSCIL, a novel framework that leverages Stable Diffusion not merely as a generative replay tool but as a unified backbone for few-shot class-incremental learning. Instead of generating and storing synthetic images, the method extracts four types of multi-scale latent features—inversion, synthetic, augmented, and generative—directly from SD’s U-Net to capture complementary semantic and structural information. These features enable efficient replay, generalization, and knowledge retention using lightweight networks (~6M parameters) while keeping SD frozen. Extensive experiments on CUB-200, miniImageNet, and CIFAR-100 benchmarks show that Diffusion-FSCIL outperforms prior FSCIL methods, demonstrating that diffusion models can act as both powerful generative and discriminative foundations for incremental learning

**Strengths:**

1. Utilizing a generative model as an encoder backbone for class incremental learning is a novel idea that has been proposed by the author.

2. There is no extra buffer space required by the model for the previously learned class for replaying the images to avoid catastrophic forgetting.

3. Using the latent features directly for replay during training is an interesting approach since the latent features captures semantics of the class rather than focusing on generating images following exact pixel distribution from data.

4. The model trained requires approximately 6M parameters.

**Weaknesses:**

1. There is a possibility of data leakage since the stable diffusion backbone used by the authors are entirely frozen which has already been trained on very large dataset.

2. In line 188, the claim is that DinoV2 is pretrained on CUB dataset, but in DinoV2 paper appendix C table 18 shows that for only for retrieval pretraining that dataset is used and not for pretraining. Which pretrained model did the authors use and if it is not retrieval pretrained then this claim doesn’t hold true and that raises the question on the claim made by the authors.

3. In Pilot Study, what is the approach used for training other models in continual setup besides Stable Diffusion Backbone. What parameters of the other models were tuned and how?

4. F_{gen} is not present during the Session 0 and introduced from session 1 (line 321) then how is it able to capture features from the classes trained in Session 0 since there is no network to distill from or use weights from. This is also highlighted in algorithm presented in Appendix where in base training F_{gen} is not present.

5. For each instance of data, forward pass generates the latent from classes for replay (F_{gen}), how is the class chosen from previous sessions for generating latent. Also, does it cover every single class that has been seen during the previous sessions?

6. Is there a particular reason why in CUB-200 there is severe degradation in other SD based models compared to the proposed approach but on miniImagenet or CIFAR 10, the degradation in performance from Session 0 to final session is almost similar.

7. For reproducibility, in line 318 what is the number of epochs that the F_{aug} is trained for.

8. For reproducibility, how is single label combination is done. What is the prompt or technique used by the authors to combine the labels into single one as claimed in line 745 of Appendix B.

9. The claim made in line 750, talks about “By employing single-word embeddings, we ensure that the semantic concept of each label is captured consistently and efficiently eliminating redundancy or semantic dilution that occurs when multi-word labels are split across multiple token.” Is there any experiment or results backing this support since it is dependent on tokenization process? For eg. in gpt-4-32k tokenizer “ring billed gull” uses 4 tokens, “Ring_billed_Gull” (used by the authors) uses 5 token and “RingBilledGull” also uses 5 tokens.

10. Some notational clarification needs to be done for eg. In line 146 S is used for denoting total number of session, instance of each incremental session.

11. \beta_{l} is used as weighing for each of the aggregation module. Are the weights being shared by each feature type or separate for each feature type.

**Questions:**

1. What is the performance of model with stable backbone on entire dataset and using similar training used for other models but with backbone of SD.

2. Clarification on how F_{gen} is distilled and used is needed. For further details see weakness point.

3. For more points refer to weakness

---

> ### Author Response · Authors · 2025-11-22
>
> > W1) There is a possibility of data leakage since the stable diffusion backbone used by the authors are entirely frozen which has already been trained on very large dataset.
>
> `Answer:` Thank you for this comment. We understand the concern regarding SD's web-scale pre-training. However, prior works [1–3] show that SD's generative objective does not produce discriminative features that perform well on vision recognition tasks without additional design. While data leakage from web-scale pre-training may exist, our pilot study in Fig. 2 indicates that simply using SD for replay does not provide effective discriminative signals for FSCIL.
>
> The improvements arise only when SD is combined with our four feature components ($\textbf{F}\_\text{inv}, \textbf{F}\_\text{syn}, \textbf{F}\_\text{gen}, \textbf{F}\_\text{aug}$) and our deliberately designed extraction pipeline, which transfers SD's generative representations into discriminative signals for FSCIL. This is further validated by the ablations in Tab. 3, where removing any feature components significantly reduces performance.
>
> We added this clarification in the revised manuscript (L77–80).
>
> **References:** \
> [1] Zhang et al., "A Tale of Two Features: Stable Diffusion Complements DINO for Zero-Shot Semantic Correspondence," NeurIPS 2023. \
> [2] Luo et al., “Diffusion Hyperfeatures: Searching Through Time and Space for Semantic Correspondence,” NeurIPS, 2023. \
> [3] Zhang et al., "Three Things We Need to Know About Transferring Stable Diffusion to Visual Dense Prediction Tasks", ECCV 2024.
>
> ---
> > W2) In line 188, the claim is that DinoV2 is pretrained on CUB dataset, but in DinoV2 paper appendix C table 18 shows that for only for retrieval pretraining that dataset is used and not for pretraining. Which pretrained model did the authors use and if it is not retrieval pretrained then this claim doesn’t hold true and that raises the question on the claim made by the authors.
>
> `Answer:` Thank you for pointing this out. We followed the interpretation used in prior work such as Diffusion Hyperfeatures [1], which described DINOv2 as being trained on “samples from CUB”. However, as pointed out by the reviewer, the official DINOv2 paper [2] clarifies in Appendix C (Table 18) that CUB-200 is used only for retrieval-based curation, not for direct pretraining. We corrected this in L189–190 and Appendix A (L728–733) of the revised manuscript.
>
> Nevertheless, we respectfully argue that the basis of our analysis remains unchanged: DINOv2's pretraining corpus explicitly incorporates a large number of CUB-related images through retrieval seeding [2]. Similarly, Rodríguez-de-Vera et al. [3] used CUB-200 for retrieval-based curation and demonstrated that such retrieval seeding significantly benefits classification performance. We think this explains DINOv2-G's higher initial accuracy. However, its sharply class-aligned discriminative features [4] are easily affected by new samples in incremental sessions, resulting in faster forgetting of previous class information when trained without auxiliary mechanisms.
>
> **References:** \
> [1] Luo et al., "Diffusion Hyperfeatures: Searching Through Time and Space for Semantic Correspondence," NeurIPS, 2023. \
> [2] Oquab et al., "DINOv2: Learning Robust Visual Features without Supervision," arXiv, 2023. \
> [3] Rodríguez-de-Vera et al., "Precision at Scale: Domain-Specific Datasets On-Demand," Pattern Recognition, 2025. \
> [4] Zhang et al., "A Tale of Two Features: Stable Diffusion Complements DINO for Zero-Shot Semantic Correspondence," NeurIPS, 2023.

---

> ### Author Response · Authors · 2025-11-22
>
> > W3. In Pilot Study, what is the approach used for training other models in continual setup besides Stable Diffusion Backbone. What parameters of the other models were tuned and how?
>
> `Answer:` We thank the reviewer for the question. In the pilot study, all comparison backbones (DINOv2, OpenCLIP, etc.) follow the same continual-learning protocol as the SD baseline. Concretely, we load the official pretrained checkpoints and freeze each backbone for all sessions. For fairness, each backbone uses identical lightweight networks (aggregation network, convolution layer, and MLP layer). During the base session, all three components are trained; in incremental sessions, only the MLP layer is trainable while others remain frozen. All backbones are trained with the NC-FSCIL [1] classifier using the DR loss under the standard FSCIL setup.
>
> For ViT-based models (DINOv2, OpenCLIP), we extract multi-layer representations from several transformer blocks and process them using the same multi-layer aggregation setup used for SD, so that all models benefit from comparable feature extraction. No model-specific finetuning or additional techniques are applied to favor any particular backbone. We added a related clarification in Appendix A (L750–755).
>
> **References:**
> [1] Yang et al., "Neural Collapse Inspired Feature–Classifier Alignment for Few-Shot Class-Incremental Learning," ICLR 2023.
>
> ---
> > W4. $\textbf{F}\_\text{gen}$ is not present during the Session 0 and introduced from session 1 (line 321) then how is it able to capture features from the classes trained in Session 0 since there is no network to distill from or use weights from. This is also highlighted in algorithm presented in Appendix where in base training F_{gen} is not present.
>
> `Answer:` Even though  $\textbf{F}\_{\text{gen}}$ is not used during the base session, two features $\textbf{F}\_{\text{inv}}$ and $\textbf{F}\_{\text{gen}}$ are extracted from the same frozen SD backbone, which accumulates knowledge through all sessions (both base and incremental sessions). Thanks to this, we are able to recover previous class information (even for base session) for $\textbf{F}\_{\text{gen}}$ in later incremental sessions. A class-specific prompt $\textbf{p}\_c^{\*}$ for each class $c$ is further required for generating $\textbf{F}\_{\text{gen}}$, which is extracted for real images at each session (both base and incremental sessions) by running "learn_prompt()", and stored throughout all incremental sessions.
>
> **Reference:** \
> [1] Gal et al., "An Image is Worth One Word: Personalizing Text-to-Image Generation using Textual Inversion," ICLR 2023.
>
> ---------------------------
> > Q2. Clarification on how $\textbf{F}\_\text{gen}$ is distilled and used is needed. For further details see weakness point.
>
> `Answer:` For distillation, the frozen $g^{\text{MLP}}$ from the previous session serves as the teacher, while the current $g^{\text{MLP}}$ acts as the student. Since $\textbf{F}\_{\text{gen}}$ is generated from noise and may exhibit generative bias relative to real-image features, we apply distillation following prior methods that reduce generative bias [1,2]: We feed $\textbf{F}\_{\text{gen}}$ to both networks and use the teacher's fixed output representation as the reference. The student is trained by mimicking the teacher's representation using $L\_\text{distill}$ described in L351.
>
> **References:** \
> [1] Smith et al., "Always Be Dreaming: A New Approach for Data-Free Class-Incremental Learning," ICCV 2021. \
> [2] Kim et al., "SDDGR: Stable diffusion-based deep generative replay for class incremental object detection," CVPR 2024.
>
> ---------------------------
> > W5. For each instance of data, forward pass generates the latent from classes for replay ($\textbf{F}\_\text{gen}$), how is the class chosen from previous sessions for generating latent. Also, does it cover every single class that has been seen during the previous sessions?
>
> `Answer:` For class selection during replay, we uniformly sample from the set of all past classes at each training iteration and generate their $\textbf{F}_{\text{gen}}$ by denoising noise conditioned on the stored prompts $\textbf{p}^{*}$. This uniform sampling ensures that all previously seen classes are covered during training.
>
> To address W4, Q2, and W5, we revised the explanation in L839–849 and Algorithm 1 (L825-830), and added citations in L348 to provide a clearer description of the procedures in the manuscript.

---

> ### Author Response · Authors · 2025-11-22
>
> > W7) For reproducibility, in line 318 what is the number of epochs that the $\mathbf{F}_{aug}$ is trained for.
>
> `Answer:` The base session is trained for 70 epochs (CUB-200), 200 epochs (miniImageNet), and 300 epochs (CIFAR-100) using $\textbf{F}\_\text{inv}$ and $\textbf{F}\_\text{syn}$. $\textbf{F}\_\text{aug}$ is then incorporated for approximately 20% of the base training duration as a fine-tuning step. We added these details in Appendix B (L776–779).
>
> ---
>
> > W8) For reproducibility, how is single label combination done? What is the prompt or technique used by the authors to combine the labels into a single one as claimed in line 745 of Appendix B?
>
> `Answer:` For each class $c$, we learn a class-specific embedding $\textbf{w}\_c^{\*}$ and its corresponding prompt $\textbf{P}\_c^{\*}$ via textual inversion [1]. Following the recommended practice in the textual inversion guidelines [2], we adopt a single-word initializer to improve synthetic quality. To obtain this canonical concept word, we used ChatGPT to rewrite each multi-word or synonym-based class label into a single representative word (e.g., "Walker hound, Walker foxhound" → "dog"). We then manually verified whether the resulting word is properly represented by the CLIP text encoder and refined it when necessary. The resulting single-word embedding is used as the initializer for $\textbf{w}\_c^{\*}$, which is further optimized through standard textual inversion training using class-specific templates (Tab. E).
>
> > W9) About the "single-word embeddings” claim and tokenizer dependence (line 750): is there any experiment supporting this, given that it depends on tokenization (e.g., 'ring billed gull' vs 'Ring_billed_Gull' vs 'RingBilledGull')?
>
> `Answer:` In our experiments, we followed the recommended practice in the textual inversion guidelines [1], using a single canonical concept word solely to improve the initialization quality of the learned embedding. Since this setting is directly inherited from the practice suggested in [1, 2], we did not perform separate experiments isolating tokenizer effects.
>
> We revised L802–837 to clearly document this process and avoid any potential misunderstanding in both W8 and W9.
>
> **Reference:** \
> [1] Hugging Face. "Textual Inversion Training.”  https://huggingface.co/docs/diffusers/en/training/text_inversion \
> [2] Gal et al., "An Image is Worth One Word: Personalizing Text-to-Image Generation using Textual Inversion,” ICLR 2023.
>
> ---
>
> > W10) Some notational clarification needs to be done for eg. In line 146 S is used for denoting total number of session, instance of each incremental session.
>
> `Answer:` Thank you for pointing this out. We revised the notation to use $i$ exclusively for the session index (L146, L147, L150, L321, L322, L351, L353, L400, L403, L780, L844), and introduced a distinct symbol $S\_{\text{total}}$ to denote the total number of sessions (L351–353). This removes any ambiguity between the two usages.
>
> ---
>
> > W11) $\beta_{l}$ is used as weighing for each of the aggregation module. Are the weights being shared by each feature type or separate for each feature type.
>
> `Answer:` The coefficients are shared across all feature types. We revised the manuscript to make this point clearer (L250) by explicitly stating that the $\beta$ parameters are shared across $\textbf{F}\_\text{gen}$, $\textbf{F}\_\text{syn}$, $\textbf{F}\_\text{inv}$, and $\textbf{F}\_\text{aug}$.
>
> To avoid confusion, we changed the duplicated notation $\beta$ used in our training loss (L351–353) to $\gamma$.
>
> ---
> > Q1. What is the performance of model with stable backbone on entire dataset and using similar training used for other models but with backbone of SD.
>
> `Answer:` We address both reviewer's two concerns below:
>
> 1. Performance on entire dataset with SD backbone: We evaluated our method when trained on the entire CUB-200 dataset (without incremental sessions) using the same frozen SD backbone. Using only the inversion feature $\textbf{F}\_\text{inv}$, our approach achieves 84.8% accuracy.
>
> 2. Performance with SD backbone using similar FSCIL training: To evaluate SD as a backbone under the similar FSCIL protocol used by other methods, we reported their performances in Tab. 1 when adapted to use our frozen SD backbone. Under this controlled comparison, our method achieves the best final accuracy (70.3% on CUB-200), outperforming both prior methods originally designed for discriminative backbones and SDDR [1], which uses SD purely as a generator for image-level replay.
>
> **Reference:** \
> [1] Jodelet et al., "Class-Incremental Learning using Diffusion Model for Distillation and Replay," ICCVW, 2023.

---

> ### Author Response · Authors · 2025-11-25
>
> > W6. Is there a particular reason why in CUB-200 there is severe degradation in other SD-based models compared to the proposed approach but on miniImageNet or CIFAR the degradation from Session 0 to the final session is almost similar?
>
> ``Answer:`` Thank you for this insightful observation. We think that the primary reason lies in the fine-grained nature of CUB-200 compared to relatively coarse-grained datasets such as miniImageNet or CIFAR-100. Under such conditions, effectively capturing subtle distinctions between closely related classes becomes crucial.
>
> Previous SD-based methods, such as SDDR [1] and DiffClass [2], face inherent challenges in resolving these subtle fine-grained class-specific details. SDDR relies on simple class-name prompts, which typically suffice for coarse-grained classes with clear identities (as in miniImageNet or CIFAR-100). However, these prompts often fail to adequately represent fine-grained classes in CUB-200, as illustrated in our Appendix (Figs. D–E), resulting in naïve synthetic images lacking discriminative characteristics. Similarly, DiffClass utilizes LoRA [3]-based fine-tuning, a method generally effective with sufficient data but inherently more challenging under the few-shot regime for fine-grained class distinctions. In contrast, our method optimizes class-specific embeddings ($\mathbf{w}^*$) to effectively capture few-shot concepts [4]; combined with knowledge distillation to reduce representational discrepancies, we think these training strategies contribute to mitigating the degradation observed on CUB-200.
>
> **References** \
> [1] Jodelet et al., "Class-Incremental Learning using Diffusion Model for Distillation and Replay," ICCVW 2023. \
> [2] Meng et al., "DiffClass: Diffusion-Based Class Incremental Learning," ECCV 2024. \
> [3] Hu et al., "LoRA: Low-Rank Adaptation of Large Language Models," ICLR 2022. \
> [4] Gal et al., "An Image is Worth One Word: Personalizing Text-to-Image Generation using Textual Inversion," ICLR 2023.

---

### Official Review · Reviewer_pnUn · 2025-10-31

**Soundness:** 2
**Presentation:** 3
**Contribution:** 2
**Rating:** 6
**Confidence:** 5

**Summary:**

This paper concentrates on Few-shot class-incremental learning (FSCIL), and proposes Diffusion-FSCIL based on Stable Diffusion (SD). Main contributions are summarized as:

a. Proposing  Diffusion-FSCIL, a framework that fully exploits SD as a unified backbone for FSCIL.

b. Extensive experiments to prove the effectiveness of proposed.

**Strengths:**

a. This paper is well written and easy to follow.

b. It is interesting to focus on Stable Diffusion in CL.

**Weaknesses:**

There are three main concerns:

a. I respectfully disagree with the motivation of this work. In section of introduction, it is mentioned that classical generative replay (GR) CL methods have the drawbacks that dependeding on separate discriminative backbones. This is not a drawback, because classical GR methods consider the situation that a customer has its own model (i.e., separate discriminative backbone). Classical GR methods aim to mitigate catastrophic forgetting of this model by adopting a generator. Therefore, this is not a drawback.

b. Instead, i think the proposed method is limited to Stable Diffusion (SD). All operations are based on SD. It is ok to propose a SD-bsed methods. But it doesn't mean that the proposed method deals with the issue of classical GR. Instead, classical GR is general to customer's backbone.

c. I think this paper is lack of technical novelty and the motivation of each operation in the proposed method, such as One-step inversion feature, One-step synthetic feature, is not clear. Why does this work use these operations? What is the motivation?

Other concern:

d. Section 5.4 reports the time cost cross incremental sessions. However, the most time cost occurs at base session.

**Questions:**

N/A

---

> ### Author Response · Authors · 2025-11-22
>
> > W1) I respectfully disagree with the motivation of this work. In section of introduction, it is mentioned that classical generative replay (GR) CL methods have the drawbacks that dependeding on separate discriminative backbones. This is not a drawback, because classical GR methods consider the situation that a customer has its own model (i.e., separate discriminative backbone). Classical GR methods aim to mitigate catastrophic forgetting of this model by adopting a generator. Therefore, this is not a drawback.
>
> `Answer:` Thank you for pointing this out. Our intention was to emphasize that our method leverages SD as a unified backbone rather than merely as a generator, offering a more comprehensive use of SD. In FSCIL, where limited samples arrive incrementally, relying on separate discriminative and generative backbones can introduce representational discrepancies that make stable knowledge retention more difficult. In contrast, using a unified backbone with augmented features—where all representations originate from the same model—yields more coherent updates and helps the model retain knowledge of previously learned classes even when new classes are introduced with very few samples. This behavior is also reflected in our experiments (Figs. 2 and 5), where naïvely applying SD-based replay does not work as expected, whereas the proposed unified-backbone design enables successful learning under the same conditions.
>
> Nonetheless, we agree with the reviewer that using a separate discriminative backbone is appropriate when a customer already has an existing model. In the revised version (L72–76), we adjusted the wording so that separate backbones are no longer described as a drawback, and we instead clarify that our approach primarily aims to avoid the additional components commonly used in SD-based GR pipelines [1,2,3,4], such as synthetic buffers, GLIGEN, or LoRA, which can increase operational complexity.
>
> **References:** \
> [1] Kim et al., “SDDGR: Stable diffusion-based deep generative replay for class incremental object detection,” CVPR 2024. \
> [2] Meng et al., “DiffClass: Diffusion-based class incremental learning,” ECCV 2024. \
> [3] Jodelet et al., “Class-incremental learning using diffusion model for distillation and replay,” ICCVW 2023. \
> [4] Jodelet et al., “Future-proofing class-incremental learning,” Machine Vision and Applications 2025.
>
> ---
> > W2) Instead, i think the proposed method is limited to Stable Diffusion (SD). All operations are based on SD. It is ok to propose a SD-bsed methods. But it doesn't mean that the proposed method deals with the issue of classical GR. Instead, classical GR is general to customer's backbone.
>
> `Answer 1:` Thank you for this constructive comment. As we clarified in our response to W1, we agree that classical GR methods are general to customer's backbone. However, our work aims to address the challenges that can arise in SD-based GR methods [1,2,3,4] by proposing a comprehensive use of SD as a unified backbone rather than merely as a generator. Indeed, our pilot study (Fig. 2 (right) in Sec. 3.2) showed that naïvely applying SD in the classical GR framework failed, which motivated us to develop a specialized framework that effectively leverages SD's rich representational capacity to improve FSCIL accuracy.
>
> Regarding generalization, our method can apply to other generative models that are sufficiently pre-trained and can extract multi-scale features that encompass both detailed patterns and higher-level semantics throughout the generative process. Recent work [5] demonstrated that SD produces such semantically rich multi-scale representations. Furthermore, other recent work [6] demonstrated that DiT architectures also produce such semantically rich multi-scale features throughout the denoising process, supporting the potential applicability of our approach to a broader range of diffusion-based generative models. To verify this, we are conducting experiments using FLUX.1 [7], a DiT-based backbone, and will update the results before the rebuttal period ends.
>
> **References:** \
> [1] Kim et al., “SDDGR: Stable diffusion-based deep generative replay for class incremental object detection,” CVPR 2024. \
> [2] Meng et al., “DiffClass: Diffusion-based class incremental learning,” ECCV 2024. \
> [3] Jodelet et al., “Class-incremental learning using diffusion model for distillation and replay,” ICCVW 2023. \
> [4] Jodelet et al., “Future-proofing class-incremental learning,” Machine Vision and Applications 2025. \
> [5] Luo et al., "Diffusion Hyperfeatures: Searching Through Time and Space for Semantic Correspondence," NeurIPS 2024. \
> [6] Gan et al., "Unleashing Diffusion Transformers for Visual Correspondence by Modulating Massive Activations," NeurIPS 2025. \
> [7] Black Forest Labs, "FLUX.1 [dev]," https://huggingface.co/black-forest-labs/FLUX.1-dev, 2024.

---

> ### Author Response · Authors · 2025-11-22
>
> > W3) I think this paper is lack of technical novelty and the motivation of each operation in the proposed method, such as One-step inversion feature, One-step synthetic feature, is not clear. Why does this work use these operations? What is the motivation?
>
> `Answer:` Thank you for this comment. We provide clarification here and adjusted the revised version accordingly. As introduced in Sec. 4.1, multi-scale diffusion features encompass both detailed patterns and higher-level semantics, which are particularly helpful for FSCIL that benefits from representations combining fine detail and semantic generalization. To effectively leverage this, we extract "one-step inversion feature" ($\textbf{F}\_\text{inv}$) without any diffusion steps, thereby capturing stable semantic information that preserves core image structure. Building upon $\textbf{F}\_\text{inv}$, our "one-step synthetic feature" ($\textbf{F}\_\text{syn}$) is naturally derived by denoising the minimally noised latent obtained from $\textbf{F}\_\text{inv}$. During this denoising process, we feed the class-name corresponding to $\textbf{F}\_\text{inv}$ into SD's text encoder to extract text-conditioned features that reflect SD's semantic prior. To reduce unnecessary overhead, we consistently adopt the one-step strategy, which is motivated by our analysis in Sec. 4.1 ("Discussion on diffusion steps") and further validated in Appendix E. We outlined the roles of each feature in Appendix G (L948–960).
>
> Technical Novelty: Our work is the first to systematically exploit both the inversion and generation pathways of state-of-the-art genersative models (i.e., SD) within a single unified backbone for FSCIL. Rather than merely using SD as an image generator, we carefully design our framework to extract complementary features from the inversion process ($\textbf{F}\_\text{inv}$) and generation processes at different noise levels ($\textbf{F}\_\text{syn}, \textbf{F}\_\text{aug}, \textbf{F}\_\text{gen}$), all within the same latent feature space. This enables seamless integration of discriminative and generative capabilities for continual learning.
>
> ---
> > W4) Section 5.4 reports the time cost cross incremental sessions. However, the most time cost occurs at base session.
>
> `Answer:` We followed a common pattern in FSCIL where most computational investment occurs during the base session to build strong representations, followed by lightweight incremental updates. After the base session, all incremental stages operate with minimal computation as the backbone remains frozen. For this reason, securing efficiency for multiple incremental sessions is generally more important than the base session complexity, which occurs only once. However, per reviewer's request, we added the base session training time in the revised manuscript (L943–945).

---

> ### Author Response · Authors · 2025-12-01
> **Regarding generalization**
>
> > W2) Instead, i think the proposed method is limited to Stable Diffusion (SD). All operations are based on SD. It is ok to propose a SD-based methods. But it doesn't mean that the proposed method deals with the issue of classical GR. Instead, classical GR is general to customer's backbone.
>
> `Answer 2:` We have completed the FLUX.1 evaluation to address the concern that our method may be limited to SD. The results show that our approach is not tied to SD's U-Net architecture and can also be applied to a DiT-based generative model. When evaluated on CUB-200 using FLUX.1 [1] as the frozen backbone, our method achieves higher accuracy than NC-FSCIL [2] across most incremental sessions:
>
> | Method   |   S0   |   S1  |   S2  |   S3  |   S4  |   S5  |   S6  |   S7  |   S8  |   S9  |  S10  |  Avg. |
> |----------|-------:|------:|------:|------:|------:|------:|------:|------:|------:|------:|------:|------:|
> | NC-FSCIL | **75.50**  | **61.22** | 54.81 | 51.66 | 49.85 | 46.21 | 44.04 | 42.56 | 39.32 | 38.51 | 37.08 | 49.16 |
> | Ours     | **75.50**  | 61.12 | **57.23** | **52.31** | **51.07** | **47.23** | **45.02** | **42.84** | **40.34** | **40.32** | **38.71** | **50.15** |
>
> Although our main framework is built around SD, these results indicate that its core principles extend beyond a single architecture, demonstrating promising potential for extension to a broader family of large-scale generative models. This evaluation was added into the revised manuscript at L994–1004 and Tab. H.
>
> **References:** \
> [1] Black Forest Labs, "FLUX.1 [dev]," https://huggingface.co/black-forest-labs/FLUX.1-dev, 2024. \
> [2] Yang et al., "Neural Collapse Inspired Feature-Classifier Alignment for Few-Shot Class-Incremental Learning," ICLR, 2023.

---

### Official Review · Reviewer_iy4T · 2025-11-01

**Soundness:** 3
**Presentation:** 3
**Contribution:** 3
**Rating:** 6
**Confidence:** 3

**Summary:**

this paper investigates the few-shot class-incremental learning (FSCIL) problem and introduces a diffusion-model-based approach. instead of only using the latent diffusion model (LDM) for generative replays, the proposed method uses LDM for feature extraction, and shows that this LDM-based feature extractor can achieve competitive performance when compared to widely adopted options like DINOv2. by integrating the clean feature, the text-conditioned one-step noised feature, and multi-step noised features, the proposed method achieves good performance on multiple benchmarks.

**Strengths:**

+ using LDM for feature extraction instead of generative replay is interesting
+ the proposed method and its usage of different clean and noised LDM features makes sense
+ ablation on the different feature components from the ovearll method
+ easy to read with clear figures

**Weaknesses:**

- the reviewer is not convinced by the claim "DINOv2-G achieves higher initial accuracy due to larger capacity and explicit pre-training on CUB-200" (L188). SD is also trained on web data (most likely including CUB dataset). plus, the proposed method needs at least four forward passes on UNet and one forward pass on VAE encoder, so most like more compute than ViT-G in DINOv2
- following the above point, please consider report the inference FLOPs cost for the overall system and compare with existing methods, since the four forward passes do seem expensive

**Questions:**

see above

---

> ### Author Response · Authors · 2025-11-22
>
> > W1-1) the reviewer is not convinced by the claim "DINOv2-G achieves higher initial accuracy due to larger capacity and explicit pre-training on CUB-200" (L188). SD is also trained on web data (most likely including CUB dataset).
>
> `Answer`: Thank you for this comment. We agree that both DINOv2 and SD are trained on large-scale web data. Nevertheless, we think there are two critical points that explain why DINOv2-G achieves higher initial accuracy than SD.
>
> First, data curation differs substantially. DINOv2 employs retrieval-based filtering for training, explicitly using datasets like CUB-200 as retrieval anchors to collect large relevant sets [1]. Prior works [2,3] have shown that such targeted curation improves downstream recognition performance. By contrast, SD is trained on broader web corpora where CUB-related content is less concentrated.
>
> Second, learning objectives are different. DINOv2 is optimized with a discriminative objective designed for recognition tasks, while SD is trained with a text-to-image reconstruction objective focused on generating diverse images from text prompts rather than learning class-separating features. Recent analyses [4,5] demonstrate that diffusion features generally underperform discriminative backbones on recognition tasks unless additional designs are introduced.
>
> We think these two differences create the initial performance gap observed in Fig. 2 (left). To improve clarity, we revised L189–190 and Appendix A (L728-733) to reorganize the description with more detailed explanations.
>
> **References:** \
> [1] Oquab et al., "DINOv2: Learning Robust Visual Features without Supervision", 2023. \
> [2] Rodríguez-de-Vera et al., "Precision at Scale: Domain-Specific Datasets On-Demand", Pattern Recognition, 2025. \
> [3] Abbas et al., "Effective Pruning of Web-Scale Datasets Based on Complexity of Concept Clusters", ICLR, 2024. \
> [4] Mukhopadhyay et al., "Do Text-Free Diffusion Models Learn Discriminative Visual Representations?", ECCV, 2024. \
> [5] Zhang et al., "Three Things We Need to Know About Transferring Stable Diffusion to Visual Dense Prediction Tasks", ECCV, 2024.
>
> ---
>
> > W1-2) The proposed method needs at least four forward passes on UNet and one forward pass on VAE encoder, so most like more compute than ViT-G in DINOv2. \
> > W2) Following the above point, please consider report the inference FLOPs cost for the overall system and compare with existing methods, since the four forward passes do seem expensive
>
> `Answer`: Thank you for this important point. During training, we obtain multiple feature types ($\textbf{F}\_\text{inv}, \textbf{F}\_\text{syn}, \textbf{F}\_\text{aug}, \textbf{F}\_\text{gen}$) through four forward passes on the SD backbone. However, at inference, the model uses only a single VAE encoding followed by one UNet forward pass to compute $\textbf{F}\_\text{inv}$.
>
> We added the inference computational cost analysis in the revised manuscript (L933–941). As the reviewer expected, the majority of our model's FLOPs arise from the frozen SD backbone, while our trainable parameters remain minimal (≈6M).

---

### Author Response · Authors · 2025-11-25

We sincerely thank all reviewers ``iy4T``, ``pnUn``, ``SWdq``, and ``7vfk`` for their insightful and constructive feedback, which significantly helped us improve our manuscript.

**Reviewers highlighted the following strengths of our work:**

- **Novel approach**: Using a generative model as an encoder backbone and frozen feature extractor for class incremental learning instead of merely as a generative replay buffer is a novel idea that is original, well-motivated, and interesting [``iy4T``, ``SWdq``, ``7vfk``]. It is interesting to focus on Stable Diffusion in continual learning [``pnUn``].
- **Technical soundness**: The proposed method and its usage of different clean and noised features makes sense [``iy4T``]. The use of multi-scale U-Net features from Stable Diffusion is technically sound and well-justified, and the class-specific prompt optimization and controlled noise injection strategies are clever and effective [``7vfk``].
- **Efficiency**: The framework eliminates the need for buffer memory to store synthetic images by using latent features directly for replay during training, which is interesting since latent features capture class semantics rather than exact pixel distributions [``SWdq``]. The model requires only approximately 6M trainable parameters [``SWdq``].
- **Presentation and evaluation**: The paper is well-written and easy to follow, with clear structure, clear figures, and intuitive illustrations [``iy4T``, ``pnUn``, ``7vfk``]. The ablation studies on different feature components demonstrate their contributions [``iy4T``].

---
**Summary of reviewers' key concerns and our responses:**

- Reviewer ``iy4T``: Requested clarification on DINOv2 pretraining and inference FLOPs. We clarified that CUB-200 was used for retrieval-based curation rather than direct pretraining, and added analysis on inference computational cost including FLOPs comparison.

- Reviewer ``pnUn``: Questioned our motivation regarding separate backbones and generalization ability beyond SD (e.g., DiT backbone), and requested clarification on the motivation and technical novelty of each operation. We clarified our motivation, conducted DiT backbone experiments to demonstrate generalizability, and better explained the distinctive roles and motivations of each feature type.

- Reviewer ``SWdq``: Requested clarification on our DINOv2 pretraining description, raised concerns about potential data leakage, and requested clarification on notations and additional implementation details (including $\textbf{F}_\text{gen}$ procedures and prompt optimization). We corrected the explanation for DINOv2 pretraining, clarified data leakage concerns and all notations, and provided comprehensive implementation details.

- Reviewer ``7vfk``: Requested experiments on larger-scale datasets and adaptation to other generative models. We added ImageNet-100 evaluation and conducted DiT backbone experiments demonstrating adaptability.

---

> ### Author Response · Authors · 2025-12-02
> **Revisions with Line Numbers**
>
> **Detailed revisions with corresponding line numbers:**
>
> - **L72–76**: We clarified our motivation by highlighting the potential challenges of traditional SD-based GR approaches; while not pointing out separate discriminative backbones as a drawback, per reviewer's comment. [``pnUn``]
> - **L77–80**: To relieve potential data leakage concerns for SD's web-scale pretraining, we cite prior works saying that auxiliary components are still required for the SD backbone to be adapted to discriminative recognition tasks. [``SWdq``]
> - **L146, L147, L150, L321, L322, L351–353, L400, L403, Appendix L780, Appendix L844**: We revised notations and symbols to remove ambiguity. [``SWdq``]
> - **L250**: We clarified that the beta weights are shared across features. [``SWdq``]
> - **L189–190, Appendix L728–733**: We corrected our earlier explanation for DINOv2 pretraining (not directly trained on CUB-200 but retrieval-curated) and supplemented additional reasons for the performance gap observed between DINOv2-G and SD in early sessions. [``iy4T``, ``SWdq``]
> - **L348**: To clarify our motivation, we added citations for prior methods that reduce generative bias through distillation. [``SWdq``]
> - **Appendix L750–755**: We added an explicit statement describing the experimental protocol for our pilot study, ensuring all backbones were evaluated under identical conditions. [``SWdq``]
> - **Appendix L776–779**: We reported the training epochs used for our training process. [``SWdq``]
> - **Appendix L802–837**: We revised details of our class-specific prompt optimization setup, clarifying the standard procedure and reorganizing the explanation regarding tokenizers. [``SWdq``]
> - **Appendix L839–849, Algorithm 1 (L825-830)**: We clarified how $\textbf{F}_\text{gen}$ is generated and used across sessions. We also updated Algorithm 1 to better describe the training procedure. [``SWdq``]
> - **Appendix L933–941**: We reported inference computational costs including FLOPs and added a detailed analysis. [``iy4T``]
> - **Appendix L943–945**: We included training time analysis for the base session to clarify the computational cost distribution. [``pnUn``]
> - **Appendix L948–960**: We summarized distinctive roles of each feature type, emphasizing their complementary roles within our unified framework. [``pnUn``, ``7vfk``]
> - **Appendix L966–993, Tab. G**: We added ImageNet-100 evaluation demonstrating strong performance on larger-scale FSCIL benchmarks. [``7vfk``]
> - **Appendix L994–1004, Tab. H**: We conducted DiT backbone experiments using FLUX.1, demonstrating that our framework can be generalized beyond SD's U-Net architecture. [``pnUn``, ``7vfk``]

---

### Meta-Review · Area_Chair_qir2 · 2025-12-28

**Summary:**

This submission received **mixed but overall borderline-to-below-threshold initial scores**, with two reviewers leaning toward acceptance (Reviewers iy4T and pnUn, both scoring 6) and two reviewers leaning toward rejection or marginal rejection (Reviewers SWdq and 7vfk, both scoring 4). While several reviewers found the idea of leveraging Stable Diffusion (SD) as a frozen feature backbone for FSCIL to be interesting, there was no clear consensus in favor of acceptance.

Across the reviews, the primary concerns centered on the practical value of the proposed design, particularly the necessity of using SD beyond its established role in generative replay. As noted by Reviewer iy4T, compared to discriminative baselines, the method incurs approximately 890 GFLOPs per image, in contrast to around 85 GFLOPs for OpenCLIP, raising serious concerns about computational efficiency and real-world applicability. Reviewer pnUn further pointed out that generative replay is well suited to scenarios where users can maintain and update their own models, calling into question whether the proposed alternative use of SD provides sufficient practical advantage.

Given the initial scores and the fact that the core concern lies in the fundamental design choice, my overall judgment as Area Chair is to recommend **rejection**.

**Reviewer Concerns:**

### Concerns partially addressed by the rebuttal

* **Clarity of motivation and explanation of feature types**:
  The authors clarified the roles of the four feature types (inversion, synthetic, augmented, generative) and added a consolidated summary explaining their complementary functions in FSCIL
  *(Reviewers pnUn, 7vfk)*.

* **Computational cost and efficiency**:
  Inference FLOPs and training-time analysis were added, clarifying that only a single UNet forward pass is required at inference and that trainable parameters remain small (~6M)
  *(Reviewer iy4T)*.

* **Generality beyond Stable Diffusion**:
  Additional experiments using a DiT-based backbone (FLUX.1) were included to suggest that the framework is not strictly tied to SD’s U-Net architecture
  *(Reviewers pnUn, 7vfk)*.

* **Reproducibility and notation issues**:
  The rebuttal substantially improved implementation details, notation consistency, and procedural clarity (e.g., prompt optimization, replay sampling, loss weights)
  *(Reviewer SWdq)*.


### Core concerns that remain outstanding

* **Questionable motivation and positioning relative to classical generative replay**:
  Reviewer pnUn explicitly disagreed with the paper’s original motivation that using separate discriminative backbones is a drawback of classical generative replay. Although the wording was softened in the revision, the paper’s positioning still does not convincingly establish that Diffusion-FSCIL addresses a limitation of existing GR methods, rather than proposing an SD-specific alternative.

* **Heavy dependence on a large pretrained diffusion backbone**:
  Reviewers raised concerns that performance gains may largely stem from SD’s web-scale pretraining rather than from the proposed FSCIL mechanisms themselves. While the authors argue that naïve SD replay fails, the risk of confounding backbone capacity with method effectiveness remains
  *(Reviewers iy4T, SWdq)*.

* **Lack of theoretical grounding**:
  Reviewer 7vfk noted the absence of a principled explanation for why diffusion features should be more robust to forgetting than discriminative features (e.g., DINOv2). The rebuttal provides intuition but does not fully resolve this gap.

**Reviewer Scores:**

* **Reviewer iy4T (initial: 6)** → **Likely unchanged slightly lower**
  While reviewers were generally positive about the idea and empirical performance, the rebuttal’s additional efficiency results ultimately reinforced the reviewers’ concerns that, compared to discriminative models, the current approach remains impractical for real-world use.

* **Reviewer pnUn (initial: 6)** → **Possibly unchanged or slightly lower**
  Maintains skepticism about motivation and novelty despite added explanations.

* **Reviewer SWdq (initial: 4)** → **Likely unchanged**
  Reproducibility issues improved, but concerns about data leakage risk, novelty, and SD dependence remain.

* **Reviewer 7vfk (initial: 4)** → **Likely unchanged**
  Acknowledges strong results but remains unconvinced by lack of theory and limited evaluation scope.

---

### Decision · Program_Chairs · 2026-01-26

Reject